# Optimization and Generalization Analysis of Transduction through Gradient Boosting and Application to Multi-scale Graph Neural Networks

**Kenta Oono**
The University of Tokyo
Preferred Networks, Inc.
Tokyo, Japan
kenta_oono@mist.i.u-tokyo.ac.jp

**Taiji Suzuki**
The University of Tokyo
RIKEN Center for Advanced Intelligence Project
Tokyo, Japan
taiji@mist.i.u-tokyo.ac.jp

## Abstract

It is known that the current graph neural networks (GNNs) are difficult to make themselves deep due to the problem known as *over-smoothing*. Multi-scale GNNs are a promising approach for mitigating the over-smoothing problem. However, there is little explanation of why it works empirically from the viewpoint of learning theory. In this study, we derive the optimization and generalization guarantees of transductive learning algorithms that include multi-scale GNNs. Using the boosting theory, we prove the convergence of the training error under weak learning-type conditions. By combining it with generalization gap bounds in terms of transductive Rademacher complexity, we show that a test error bound of a specific type of multi-scale GNNs that decreases corresponding to the number of node aggregations under some conditions. Our results offer theoretical explanations for the effectiveness of the multi-scale structure against the over-smoothing problem. We apply boosting algorithms to the training of multi-scale GNNs for real-world node prediction tasks. We confirm that its performance is comparable to existing GNNs, and the practical behaviors are consistent with theoretical observations. Code is available at https://github.com/delta2323/GB-GNN.

## 1 Introduction

Graph neural networks (GNNs) [27, 52] are an emerging deep learning model for analyzing graph structured-data. They have achieved state-of-the-art performances in node prediction tasks on a graph in various fields such as biochemistry [17], computer vision [69], and knowledge graph analysis [57]. While they are promising, the current design of GNNs has witnessed a challenge known as *over-smoothing* [38, 48]. Typically, a GNN iteratively aggregates and mixes node representations of a graph [26, 36, 64]. Although it can capture the subgraph information using local operations only, it smoothens the representations and makes them become indistinguishable (over-smoothen) between nodes as we stack too many layers, leading to underfitting of the model. Several studies suspected that this is the cause of the performance degradation of deep GNNs and devised methods to mitigate it [51, 70]. Among others, multi-scale GNNs [39, 45, 68] are a promising approach as a solution for the over-smoothing problem. These models are designed to combine the subgraph information at various scales, for example, by bypassing the output of the middle layers of a GNN to the final layer.

Although multi-scale GNNs empirically have resolved the over-smoothing problem to some extent, little is known how it works theoretically. To justify the empirical performance from the viewpoint of statistical learning theory, we need to analyze two factors: *generalization gap* and *optimization*. There are several studies to guarantee the generalization gaps [15, 24, 33, 53, 65]. However, to the best of our knowledge, few studies have provided optimization guarantees. The difficulty partly

originates owing to the inter-dependency of predictions. That is, the prediction for a node depends on the neighboring nodes, as well as its feature vector. It prevents us from extending the optimization theory for inductive learning settings to transductive ones.

In this study, we propose the analysis of multi-scale GNNs through the lens of the boosting theory [31, 46]. Our idea is to separate a model into two types of functions – aggregation functions $\mathcal{G}$ that mix the representations of nodes and transformation functions $\mathcal{B}$, typically common to all nodes, that convert the representations to predictions. Accordingly, we can interpret a multi-scale GNN as an ensemble of supervised models and incorporate analysis tools of inductive settings. We first consider our model in full generality and prove that as long as the model satisfies the *weak learning condition* (w.l.c.), which is a standard type of assumption in the boosting theory, it converges to the global optimum. By combining it with the evaluation of the transductive version of Rademacher complexity [19], we give a sufficient condition under which a particular type of multi-scale GNNs has the upper bound of test errors that decreases with respect to depth (the number of node aggregation operations) under the w.l.c. This is in contrast to usual GNNs suffering from the over-smoothing problem. Finally, we apply multi-scale GNNs trained with boosting algorithms, termed *Gradient Boosting Graph Neural Network* (GB-GNN), to node prediction tasks on standard benchmark datasets. We confirm that our algorithm can perform favorably compared with state-of-the-art GNNs, and our theoretical observations are consistent with the practical behaviors.

The contributions of this study can be summarized as follows:

- We propose the analysis of transductive learning models via the boosting theory and derive the optimization and generalization guarantees under the w.l.c. (Theorem 1, Proposition 2).

- As a special case, we give the test error bound of a particular type of multi-scale GNNs that monotonically decreases with respect to the number of node aggregations (Theorem 2).

- We apply GB-GNNs, GNNs trained with boosting algorithms, to node prediction tasks on real-world datasets. We confirm that GB-GNNs perform favorably compared with state-of-the-art GNNs, and theoretical observations are consistent with the empirical behaviors.

**Notation** $\mathbb{N}_+$ denotes the set of non-negative integers. For $N \in \mathbb{N}_+$, we define $[N] := \{1, \ldots, N\}$. For a proposition $P$, $\mathbf{1}\{P\}$ equals $1$ when $P$ is true and $0$ otherwise. For $a, b \in \mathbb{R}$, we denote $a \wedge b := \min(a, b)$ and $a \vee b := \max(a, b)$. For a vector $u, v \in \mathbb{R}^N$ and $p \geq 1$, we denote the $p$-norm of $v$ by $\|v\|_p^p := \sum_{n=1}^N v_n^p$ and the Kronecker product by $u \otimes v$. All vectors are column vectors. For a matrix $X, Y \in \mathbb{R}^{N \times C}$ and $p \geq 1$, we define the inner product by $\langle X, Y \rangle := \sum_{n=1}^N \sum_{c=1}^C X_{nc} Y_{nc}$, $(2, p)$-norm of $X$ by $\|X\|_{2,p}^p := \sum_{c=1}^C \left( \sum_{n=1}^N X_{nc}^2 \right)^{\frac{p}{2}}$, the Frobenius norm by $\|X\|_{\mathrm{F}} := \|X\|_{2,2}$, and the operator norm by $\|X\|_{\mathrm{op}} := \sup_{v \in \mathbb{R}^C, \|v\|_2 = 1} \|Xv\|_2$. For $y \in \{0, 1\}$, we write $y^\sharp := 2y - 1 \in \{\pm 1\}$. For $a \in \mathbb{R}$, we define $\mathrm{sign}(a) = 1$ if $a \geq 0$ and $-1$ otherwise.

## 2  Related Work

**Graph-based Transductive Learning Algorithms**  Graph-based transductive learning algorithms operate on a graph given *a priori* or constructed from the representations of samples. For example, spectral graph transducer [34] and the algorithm proposed in [5] considered the regularization defined by the graph Laplacian. Another example is the label propagation algorithm [71], which propagates label information through a graph. The extension of label propagation to deep models achieved state-of-the-art prediction accuracy in semi-supervised tasks appeared in computer vision [32]. Recently, GNNs [27, 52] have been used to solve node prediction problems as a transductive learning task, where each sample point is represented as a node on a graph, and the goal is to predict the properties of the nodes. GNNs, especially MPNN-type (message passing neural networks) GNNs [26], differ from the aforementioned classical transductive learning algorithms because it mixes representations of sample points directly thorough the underlying graph.

**Over-smoothing and Multi-scale GNNs**  Multi-scale GNNs [1, 2, 7, 39, 40, 45, 68] are a promising approach for mitigating the over-smoothing problem using the information of subgraphs at various scales. For example, the Jumping Knowledge Network [68] were intentionally designed to solve the over-smoothing problem by aggregating the outputs of the intermediate layers to the final layer.

However, to the best of our knowledge, there is no theoretical explanation of why multi-scale GNNs can perform well against the over-smoothing problem. We proved that a specific instantiation of our model has a test error bound that monotonically decreases with respect to depth, thereby providing the evidence for the architectural superiority of multi-scale GNNs for the over-smoothing problem.

**Boosting Interpretation of Deep Models**  Boosting [20, 54] is a type of ensemble method for combining several learners to create a more accurate one. For example, gradient boosting [22, 41] is a de-facto boosting algorithm owing to its superior practical performance and easy-to-use libraries [10, 35, 50]. Reference [63] interpreted Residual Network (ResNet) [30] as a collection of relatively shallow networks. References [31, 46] gave another interpretation as an ensemble model and evaluated its theoretical optimization and generalization performance. In particular, [46] employed the notion of (functional) gradient boosting. Similar to these studies, we interpret a GNN as an ensemble model to derive the optimization and generalization guarantees.

AdaGCN (AdaBoosting graph convolutional network), which has been recently proposed by [59], is the closest to our study. They interpreted a multi-scale GCN (graph convolutional network) [36] as an ensemble model and trained it using AdaBoost [21]. Although their research demonstrated the practical superiority of the boosting approach, we would argue that there is room for exploration in their theory. For example, they used the Vapnik—Chervonenkis (VC) dimension to evaluate the generalization gap. However, it is known that the VC dimension cannot explain the empirical behaviors of AdaBoost [55] (see also [43, Section 7.3]). Besides, they did not give optimization guarantees of AdaGCNs. In contrast, our primary goal is to devise methodologies for multi-scale GNNs with a solid theoretical backbone. To realize it, we tackle the non-i.i.d. nature of node prediction tasks and derive the optimization and refined generalization guarantees.

**Generalization Analysis of GNNs in Transductive Settings**  It is not trivial to define the appropriate notion of generalization in a transductive learning setting because we do not need to consider the prediction accuracy of sample points that are not in a given dataset. We define the generalization gap as of discrepancy between the training and test errors in terms of the random partition of a full dataset into training and test datasets[19, 62] (see Section 4.2 for the precise definition). This definition can admit the dependency between sample points. Furthermore, [49] showed that any generalization gap bound in this setting is automatically translated to the bound of the corresponding i.i.d. setting. We employed the transductive version of Rademacher complexity, introduced by [19] to bound generalization gaps. Similarly to supervised settings, we have the transductive version of model complexities such as the VC dimension and variants of Rademacher complexity [60, 61]. We also have transductive PAC-Bayes bounds [8] and stability-based bounds [12, 18] for generalization analysis. Although several research have studied generalization of GNNs [15, 24, 33, 53, 65], to the best of our knowledge, none of them satisfies for our purpose (see Section B).

## 3  Problem Settings

**Transductive Learning**  Let $\mathcal{X}$ and $\mathcal{Y}$ be spaces of feature vectors and labels, respectively. Let $N \in \mathbb{N}_+$ be the sample size and $V := [N]$ be the set of indices of the sample. For each sample point $i \in V$, we associate a feature-label pair $(x_i, y_i) \in \mathcal{X} \times \mathcal{Y}$. Let $V_{\text{train}}$ and $V_{\text{test}}$ be the set of training and test samples, respectively, satisfying $V_{\text{train}} \cap V_{\text{test}} = \emptyset$ and $V_{\text{train}} \cup V_{\text{test}} = V$. We denote the training and test sample sizes by $M := |V_{\text{train}}|$ and $U := |V_{\text{test}}|$, respectively. Given the collection of features $X = (x_i)_{i \in V}$ and labels $(y_i)_{i \in V_{\text{train}}}$ for the training data, the task is to construct a predictor $h : \mathcal{X} \to \widehat{\mathcal{Y}}$ such that $h(x_i)$ is *close* to $y_i$ for all $i \in V_{\text{test}}$ (we define it precisely in Section 4.1). Here, $\widehat{\mathcal{Y}}$ denotes the range of the predictor. For later use, we define $Q := \frac{1}{M} + \frac{1}{U}$.

**Gradient Boosting**  We briefly present an overview of the gradient boosting method [23, 41], which is also called the restricted gradient descent [28]. Let $\mathcal{H}$ be a subset of Hilbert space (e.g., a collection of predictors). Given a functional $\mathcal{L} : \mathcal{H} \to \mathbb{R}$ (e.g., training error), we want to find the minima of $\mathcal{L}$. Gradient boosting solves this problem by iteratively updating the predictor $h^{(t)} \in \mathcal{H}$ at each iteration $t$ by adding a weak learner $f^{(t)}$ near the steepest direction of $\mathcal{L}$. Although a general theory can admit that $\mathcal{H}$ is infinite-dimensional (known as *functional* gradient boosting), it is sufficient for our purpose to assume that $\mathcal{H}$ is finite-dimensional. Let $\mathcal{F}^{(t)} \subset \mathcal{H}$ a hypothesis space of weak learners at iteration $t \in \mathbb{N}_+$. Gradient boosting attempts to find $f^{(t)} \in \mathcal{F}^{(t)}$ such

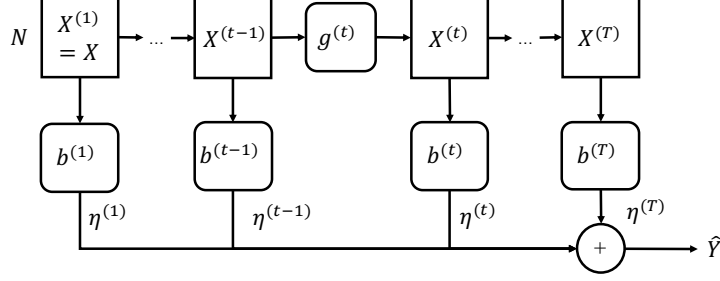

Figure 1: Schematic view of the model. $g^{(t)} : \mathcal{X}^N \to \mathcal{X}^N \in \mathcal{G}^{(t)}$ and $b^{(t)} : \mathcal{X}^N \to \widehat{\mathcal{Y}}^N \in \mathcal{B}^{(t)}$ are aggregation and transformation functions, respectively, and $\eta^{(t)}$ is the learning rate at the $t$-th iteration. We assume $\widehat{\mathcal{Y}} = \mathbb{R}$ in Sections 4 and 5 and $\mathcal{X} = \mathbb{R}^C$ in Section 5.

that $f^{(t)} \in \arg\min_{f \in \mathcal{F}^{(t)}} d(-\nabla\mathcal{L}(h^{(t)}), f)$ holds true, and the step size $\eta^{(t)} > 0$. Here, $d$ is some distance on $\mathcal{H}$ and $\nabla\mathcal{L}(h)$ is the (Fréchet) derivative of $\mathcal{L}$ at $h$. We update the predictor by $h^{(t+1)} = h^{(t)} + \eta^{(t)} f^{(t)}$. Because we cannot solve the minimization problem above exactly in most cases, we resort to an approximated algorithm that can find the solution near the optimal one (corresponding to Definition 1 below in our setting). Several boosting algorithms such as AdaBoost, Arc-x4 [6], Confidence Boost [56], and Logit Boost [22] fall into this formulation by appropriately selecting $\mathcal{L}$, $d$ and $\eta^{(t)}$'s [41].

**Models**  Figure 1 shows a schematic view of the model considered in this study. It consists of two types of components: *aggregation* functions $g^{(t)} : \mathcal{X}^N \to \mathcal{X}^N$ that mix the representations of sample points and *transformation* functions $b^{(t)} : \mathcal{X}^N \to \widehat{\mathcal{Y}}^N$ that make predictions from representations. We specify a model by defining the set of aggregation and transformation functions at each iteration $t$, denoted by $\mathcal{G}^{(t)}$ and $\mathcal{B}^{(t)}$, respectively. If we use the same function classes $\mathcal{G}^{(t)}$ and $\mathcal{B}^{(t)}$ for all $t$, we shall omit the superscript $(t)$. Typically, a transformation function $b \in \mathcal{B}^{(t)}$ is a broadcast of the same function, i.e., $b$ is of the form $b = (b_0, \ldots, b_0)$ for some $b_0 : \mathcal{X} \to \widehat{\mathcal{Y}}$. However, we do not assume this until necessary. We define the hypothesis space $\mathcal{F}^{(t)}$ at the $t$-th iteration by $\mathcal{F}^{(t)} := \{b^{(t)} \circ g^{(t)} \circ \cdots \circ g^{(1)} \mid b^{(t)} \in \mathcal{B}^{(t)}, g^{(s)} \in \mathcal{G}^{(s)} (s \in [t])\}$. Given $g^{(s)} \in \mathcal{G}^{(s)}$ selected at the $s = 1, \ldots, t-1$ iterations, we choose $g^{(t)} \in \mathcal{G}^{(t)}$ and $b^{(t)} \in \mathcal{B}^{(t)}$ to construct a weak learner $f^{(t)}(X) := b^{(t)}(g^{(t)}(X^{(t-1)}))$ and update the representation $X^{(t)} := g^{(t)}(X^{(t-1)})$. When $t = 1$, we define $f^{(1)}(X) := b^{(1)}(X)$ and $X^{(1)} := X$. We do not select $g^{(1)}$, nor do we update the representation. Algorithm 1 shows the overall training algorithm.

## 4  Analysis

### 4.1  Optimization

In this section, we focus on a binary classification problem. Accordingly, we set $\mathcal{Y} = \{0, 1\}$ and $\widehat{\mathcal{Y}} = \mathbb{R}$. For $\delta \geq 0$, we define $\ell_\delta(\hat{y}, y) := \mathbf{1}[(2p-1)y^\sharp < \delta]$, where $p = \text{sigmoid}(\hat{y}) = (1 + \exp(-\hat{y}))^{-1}$. Note that $\ell_{\delta=0}$ is the 0–1 loss. Because it is difficult to optimize $\ell_\delta$, we define the sigmoid cross entropy loss $\ell_\sigma(\hat{y}, y) := -y \log p - (1-y) \log(1-p)$ as a surrogate function. For a predictor $h : \mathcal{X} \to \widehat{\mathcal{Y}}$, we define the test error by $\mathcal{R}(h) := \frac{1}{U} \sum_{n \in V_{\text{test}}} \ell_\delta(h(x_n), y_n)$, and training errors by $\widehat{\mathcal{R}}(h) := \frac{1}{M} \sum_{n \in V_{\text{train}}} \ell_\delta(h(x_n), y_n)$ and $\widehat{\mathcal{L}}(h) := \frac{1}{M} \sum_{n \in V_{\text{train}}} \ell_\sigma(h(x_n), y_n)$. Because it is sufficient to make predictions of given samples, the values of a predictor outside of the samples do not affect the problem. Therefore, we can and do identify a predictor $h$ with a vector $\widehat{Y} := (h(x_1), \ldots, h(x_N))^\top \in \widehat{\mathcal{Y}}^N$. Accordingly, we represent $\mathcal{R}(\widehat{Y}) := \mathcal{R}(h)$ (same is true for other errors). Similarly to previous studies [28, 46], we assume the following learnability condition to obtain the optimization guarantee.

**Definition 1** (Weak Learning Condition). *Let $\alpha > \beta \geq 0$, and $\boldsymbol{g} \in \mathbb{R}^N$, we say $Z \in \mathbb{R}^N$ satisfies $(\alpha, \beta, \boldsymbol{g})$-weak learning condition (w.l.c.) if it satisfies $\|Z - \alpha\boldsymbol{g}\|_2 \leq \beta\|\boldsymbol{g}\|_2$. We say a weak learner $f : \mathcal{X}^N \to \widehat{\mathcal{Y}}^N = \mathbb{R}^N$ satisfies $(\alpha, \beta, \boldsymbol{g})$-w.l.c. when $f(X)$ does.*

---

**Algorithm 1** Training Algorithm

---

**input** Features $X \in \mathcal{X}^N$. Labels $y_i \in \mathcal{Y}$ ($i \in V_{\text{train}}$). #iterations $T$. w.l.c. params $(\alpha_t, \beta_t)_{t \in [T]}$.

**output** A collection of predictions $\widehat{Y} \in \widehat{\mathcal{Y}}^N$ for all sample points.

    Find $b^{(1)} \in \mathcal{B}^{(1)}$ and $\eta^{(1)} > 0$.

    $X^{(1)} \leftarrow X$.

    $\widehat{Y}^{(1)} \leftarrow \eta^{(1)} b^{(1)}(X)$.

    **for** $t = 2 \ldots T + 1$ **do**

        Find $b^{(t)} \in \mathcal{B}^{(t)}$ and $g^{(t)} \in \mathcal{G}^{(t)}$ and set $f^{(t)}(X) \leftarrow b^{(t)}(g^{(t)}(X^{(t-1)}))$.

        Ensure $\frac{1}{M} f^{(t)}$ satisfies $(\alpha_t, \beta_t, -\nabla \widehat{\mathcal{L}}(\widehat{Y}^{(t-1)}))$-w.l.c.

        $X^{(t)} \leftarrow g^{(t)}(X^{(t-1)})$.

        $\widehat{Y}^{(t)} \leftarrow \widehat{Y}^{(t-1)} + \eta^{(t)} f^{(t)}(X)$.

    **end for**

    $t^* = \arg\min_{t \in [T-1]} \|\nabla \widehat{L}(\widehat{Y}^{(t)})\|_{2,1}$.

    $\widehat{Y} \leftarrow \widehat{Y}^{(t^*)}$.

---

The following proposition provides a handy way to check the empirical satisfiability of w.l.c. It ensures that the weak learner and negative gradient face the same "direction". Using the arugment similar to [28], we show that our w.l.c. is equivalent to the AdaBoost-style learnability condition [31].

**Proposition 1.** *Let $Z, g \in \mathbb{R}^N$ such that $g \neq 0$. There exists $\alpha > \beta \geq 0$ such that $Z$ satisfies $(\alpha, \beta, g)$-w.l.c. if and only if $\langle Z, g \rangle > 0$. Further, when $Z \in \{\pm 1\}^N$, this is equivalent to the condition that there exists $\delta \in (0, 1]$ such that $\sum_{n=1}^{N} w_n \mathbf{1}\{\text{sign}(g_n) \neq Z_n\} \leq \frac{1-\delta}{2}$ where $w_n = \frac{g_n}{\|g\|_1}$.*

See Section A.1 for the proof. Under the condition, we have the following optimization guarantee.

**Theorem 1.** *Let $T \in \mathbb{N}_+$ and $\alpha_t > \beta_t \geq 0$ ($t \in [T]$). Define $\gamma_t := \frac{\alpha_t^2 - \beta_t^2}{\alpha_t^2}$ and $\Gamma_T := \sum_{t=1}^{T} \gamma_t$. If Algorithm 1 finds a weak learner $f^{(t)}$ for any $t \in [T]$, its output $\widehat{Y} \in \widehat{\mathcal{Y}}^N$ satisfies $\widehat{\mathcal{R}}(\widehat{Y}) \leq \frac{(1+e^\delta)\widehat{\mathcal{L}}(\widehat{Y}^{(1)})}{2M\Gamma_T}$. In particular, when $\gamma_t$ is independent of $t$, the right hand side is $O(1/T)$.*

Note that $\gamma_t$ is the lower bound of the cosine value of the angle between $f^{(t)}(X)$ and $-\nabla \widehat{\mathcal{L}}(\widehat{Y})$. The proof strategy is similar to that of [46, Theorem 1] in that we bound the gradient of the training loss (Lemma 1) and apply a Kurdyka-Łojasiewicz-like inequality (Lemma 2). See Section A.2 for the proof. We shall confirm that the w.l.c. holds empirically in the experiments in Section 7 and discuss the provable satisfiability of the w.l.c. in Section 8.

### 4.2 Generalization

We follow the problem setting of [19]. For fixed $M \in \mathbb{N}_+$, we create a training set by uniformly randomly drawing $M$ sample points *without* replacement from $V$ and treating the remaining $U$ sample points as a test set. We think of training and test errors as random variables with respect to the random partition of $V$. Reference [19] introduced the Rademacher complexity for transductive learning and derived the generalization gap bounds. We obtain the following proposition by applying it to our setting. For a hypothesis space $\mathcal{F} \subset \{\mathcal{X} \to \widehat{\mathcal{Y}}\}$, we denote its transductive Rademacher complexity by $\mathfrak{R}(\mathcal{F})$. We define $S := \frac{4(M+U)(M \wedge U)}{(2(M+U)-1)(2(M \wedge U)-1)}$, which is close to 1 when $M$ and $U$ are sufficiently large. See Section A.3 for the definition of $\mathfrak{R}(\cdot)$ and the proof of the proposition.

**Proposition 2.** *There exists a universal constant $c_0 > 0$ such that for any $\delta' > 0$, with a probability of at least $1 - \delta'$ over the random partition of samples, the output $\hat{Y}$ of Algorithm 1 satisfies*

$$\mathcal{R}(\widehat{Y}) \leq \widehat{\mathcal{R}}(\widehat{Y}) + \sum_{t=1}^{T} \eta^{(t)} \mathfrak{R}(\mathcal{F}^{(t)}) + c_0 Q \sqrt{M \wedge U} + \sqrt{\frac{SQ}{2} \log \frac{1}{\delta'}}.$$

## 5 Application to Multi-scale GNNs

We shall specialize our model and derive a test error bound for multi-scale GNNs that is monotonically decreasing with respect to $T$. In later sections, we assume $\mathcal{X} = \mathbb{R}^C$ for some $C \in \mathbb{N}_+$. We continue

to assume that $\mathcal{Y} = \{0, 1\}$ and $\widehat{\mathcal{Y}} = \mathbb{R}$. First, we specialize $\mathcal{B}^{(t)}$ as a parallel application of the same transformation function for a single sample point of the form

$$\mathcal{B}^{(t)} := \{(f_{\text{base}}, \dots f_{\text{base}})^\top \mid f_{\text{base}} \in \mathcal{B}^{(t)}_{\text{base}}\} \tag{1}$$

for some $\mathcal{B}^{(t)}_{\text{base}} \subset \{\mathcal{X} \to \widehat{\mathcal{Y}}\}$. By assuming this structure, we can evaluate the transductive Rademacher complexity in a similar way to the inductive case. We take multi layer perceptrons (MLPs) as base functions for an example (see Proposition 4 for the general case). Let $L \in \mathbb{N}_+$ $\boldsymbol{C} = (C_1, \dots, C_{L+1}) \in \mathbb{N}_+^{L+1}$ and $B > 0$ such that $C_1 = C$ and $C_{L+1} = 1$. We define $\mathcal{B}^{(t)}_{\text{base}} = \mathcal{B}^{(t)}_{\text{base}}(\boldsymbol{C}, L, \tilde{B}^{(t)}, \sigma)$ as a collection of $L$-layered MLPs with width $\boldsymbol{C}$:

$$\mathcal{B}^{(t)}_{\text{base}} := \left\{ \boldsymbol{x} \mapsto \sigma(\cdots \sigma(\boldsymbol{x}W^{(1)}) \cdots W^{(L-1)})W^{(L)} \mid \|W_{\cdot c}^{(l)}\|_1 \le \tilde{B}^{(t)} \text{ for all } c \in [H_{l+1}] \right\}. \tag{2}$$

Here, $W^{(l)} \in \mathbb{R}^{C_l \times C_{l+1}}$ for $l = 1, \dots, L^1$ and $\sigma : \mathbb{R} \to \mathbb{R}$ is a 1-Lipschitz function such that $\sigma(0) = 0$ (e.g., ReLU and sigmoid). We apply $\sigma$ to a vector in an element-wise manner. For $t \in \mathbb{N}_+$, $\tilde{P}^{(t)} \in \mathbb{R}^{N \times N}$, and $\tilde{C}^{(t)} > 0$, we use the aggregation functions $\mathcal{G}^{(t)} = \mathcal{G}^{(t)}(\tilde{P}^{(t)}, \tilde{C}^{(t)})$ defined by

$$\mathcal{G}^{(t)} := \{X \mapsto \tilde{P}^{(t)}XW \mid W \in \mathbb{R}^{C \times C} \mid \|W_{\cdot c}\|_1 \le \tilde{C}^{(t)} \text{ for all } c \in [C]\}. \tag{3}$$

When we have a graph $G$ whose nodes are identified with sample points, typical choices of $\tilde{P}^{(t)}$ are the (normalized) adjacency matrix $A$ of $G$, its augmented variant $\tilde{A}$ used in GCN[2], (normalized) graph Laplacian, or their polynomial used in e.g., LanczosNet [39]. We can evaluate Rademacher complexity as follows. By combining it with Propositions 2 and Theorem 1, we obtain test error bounds for multi-scale GNNs. See Sections A.4 and A.5 for the proof.

**Proposition 3.** *Suppose we use $\mathcal{B}^{(t)}$ and $\mathcal{G}^{(t)}$ defined above. Let $D^{(t)} = 2\sqrt{2}(2\tilde{B}^{(t)})^{L-1} \prod_{s=2}^t \tilde{C}^{(s)}$ and $P^{(t)} := \prod_{s=2}^t \tilde{P}^{(s)}$. We have $\mathfrak{R}(\mathcal{F}^{(t)}) \le \frac{1}{\sqrt{MU}} D^{(t)} \|P^{(t)}X\|_{\text{F}}$.*

**Theorem 2.** *Suppose we use $\mathcal{B}^{(t)}$ and $\mathcal{G}^{(t)}$ defined above. Let $T \in \mathbb{N}_+$ and $\alpha_t > \beta_t \ge 0$ ($t \in [T]$). Suppose Algorithm 1 with the learning rate $\eta^{(t)} = \frac{4}{\alpha_t}$ finds a weak learner $f^{(t)}$ for any $t \in [T]$. Then, for any $\delta' > 0$, with a probability of at least $1 - \delta'$, its output satisfies*

$$\mathcal{R}(\widehat{Y}) \le \frac{(1 + e^\delta)\widehat{\mathcal{L}}(\widehat{Y}^{(1)})}{2M\Gamma_T} + \frac{4}{\sqrt{MU}} \sum_{t=1}^T \frac{D^{(t)}\|P^{(t)}X\|_{\text{F}}}{\alpha_t} + c_0 Q\sqrt{M \wedge U} + \sqrt{\frac{SQ}{2}\log\frac{1}{\delta'}}. \tag{4}$$

*In particular, if $\Gamma_T = \Omega(T^\varepsilon)$ for some $\varepsilon > 0$, the first term is asymptotically monotonically decreasing with respect to $T$. If $\alpha_t^{-1} D^{(t)} \|P^{(t)}\|_{\text{op}} = O(\tilde{\varepsilon}^t)$ for some $\tilde{\varepsilon} \in (0, 1)$ independent of $T$, the second term is bounded by a constant independent of $T$.*

**Analogy to AdaBoost Bounds** By interpreting $T$ as the depth of a GNN, this theorem clarifies how the information of intermediate layers helps to mitigate the over-smoothing problem, that is, *the deeper a model is, the better it is*. Theorem 2 is similar to typical test error bounds for AdaBoost in that it consists of monotonically decreasing training error terms and model complexity terms independent of $T$ (e.g., [43, Corollay 7.5]). While hypothesis spaces are fixed for all iterations $t$ in the AdaBoost case, they can vary in our case due to the representation mixing caused by $\mathcal{G}^{(t)}$'s. The condition on $\|P\|_{\text{op}}$ ensures that the hypothesis space does not grow significantly.

**Trade-off between Model Complexity and Weak Learning Condition** There is a trade-off between the model complexity and the satisfiability of the w.l.c. Suppose we use the normalized adjacency matrix $A$ as $\tilde{P}^{(t)}$ for all $t$ (we can alternatively use the augmented version $\tilde{A}$). If the underlying graph is connected and non-bipartite, then, it is known that the eigenvalues of $A$ satisfies $1 = \lambda_1 > \lambda_2 \ge \cdots \ge \lambda_N > -1$ (e.g., [11, Lemma 1.7], [48, Proposition 1]). Let $(\xi_n)_{n \in [N]}$ be the orthonormal basis consisting of eigenvectors of $A$ and we decompose $X$ as $X_c = \sum_{n=1}^N a_{nc}\xi_n$ ($a_{nc} \in \mathbb{R}$). We denote $\tilde{X}^{(t)} := P^{(t)}X$. Assume that $|\lambda_n|$'s are small for $n \ge 2$. On the one hand, we have $\|\tilde{X}^{(t)}\|_{\text{F}}^2 = \|X\|_{\text{F}}^2 - \sum_{n \ge 2}\sum_{c=1}^C(1 - \lambda_n^{2t})a_{nc}^2$. Therefore, the model complexity terms

decrease rapidly with respect to $t$. On the other hand, we have $\tilde{X}_c^{(t)} = \xi_{1c} a_{1c} + \sum_{n=2}^N \lambda_n^t \xi_n a_{nc}$. Therefore, $\tilde{X}^{(t)}$ degenerates to a rank-one vector of the form $\xi_1 \otimes v$ ($v \in \mathbb{R}^C$) quickly under the condition. Since it is known that $(\xi_1)_i \propto \deg(i)^{\frac{1}{2}}$ where $\deg(\cdot)$ is the node degree (e.g., see [11]), $\tilde{X}^{(t)}$ has little information for distinguishing nodes other than node degrees (corresponding to the information-less space $\mathcal{M}$ in [48]). Therefore, it is hard for weak learners to satisfy w.l.c. using the smoothened representations $\tilde{X}^{(t)}$. We shall discuss the large model complexity case in Section 8.

**General Transformation Functions**  We have used MLPs as a specific choice of $\mathcal{B}^{(t)}$. More generally, by using the proposition below, we can reduce the computation of the transductive Rademacher complexity to that of the inductive counterpart without any structural assumption on $\mathcal{B}^{(t)}$ other than the parallel function application of the form Equation (1). See Section A.6 for the proof. Note that if the order of training and test sample sizes are same, this bound does not worsen the dependency on sample sizes. This assumption corresponds to the case where the ratio $r$ defined below satisfies $r = \Theta(1)$ as a function of $N$.

**Proposition 4.**  *Let $r := \frac{U}{M}$. Suppose $\mathcal{B}^{(t)}$ is of the form Equation (1) for some $\mathcal{B}_{\text{base}}^{(t)}$. Use Equation (3) as $\mathcal{G}^{(s)}$ for $s \in [t]$. Define $\mathcal{F}_{\text{base}}^{(t)} \subset \{\mathcal{X} \to \widehat{\mathcal{Y}}\}$ by*

$$\mathcal{F}_{\text{base}}^{(t)} := \{\boldsymbol{x} \mapsto f(\boldsymbol{x} W^{(2)} \cdots W^{(t)}) \mid f \in \mathcal{B}_{\text{base}}^{(t)}, \|W_{c\cdot}^{(s)}\|_1 \leq \tilde{C}^{(s)}, \forall c \in [C], s = 2, \ldots, t\}.$$

*We denote the (non-transductive) empirical Rademacher complexity of $\mathcal{F}_{\text{base}}^{(t)}$ conditioned on $P^{(t)}X$ by $\widehat{\mathfrak{R}}_{\text{ind}}(\mathcal{F}_{\text{base}}^{(t)}; P^{(t)}X)$ (see Definition 5). Then, we have $\mathfrak{R}(\mathcal{F}^{(t)}) < \frac{(1+r)^2}{r} \widehat{\mathfrak{R}}_{\text{ind}}(\mathcal{F}_{\text{base}}^{(t)}; P^{(t)}X).$*

## 6  Practical Considerations

**Learning Kernels**  The one-layer transformation of a GNN $X \mapsto AXW$ can be considered as a kernelized linear model whose Gram matrix is $\mathcal{K} = AXX^\top A$. This interpretation motivates us to select aggregate functions by learning appropriate kernels from data. We employ kernel target alignment (KTA) [13, 14] typically used in the context of kernel methods. Specifically, for $Z \in \mathbb{R}^{N \times C}$, we denote its Gram matrix $\mathcal{K}[Z] \in \mathbb{R}^{M \times M}$ of the training data by $\mathcal{K}[Z]_{ij} := Z_i Z_j^\top$ for $i, j \in V_{\text{train}}$. We define the correlation $\rho$ by $\rho(Z, Z') := \frac{\langle \mathcal{K}[Z], \mathcal{K}[Z'] \rangle}{\|\mathcal{K}[Z]\|_{\text{F}} \|\mathcal{K}[Z']\|_{\text{F}}}$. Given a set of aggregation functions $\mathcal{G}_{\text{KTA}}^{(t)}$, we choose the aggregation function $g^{(t)}$ such that $g^{(t)} \in \arg\max_{g \in \mathcal{G}_{\text{KTA}}^{(t)}} \rho(g(X^{(t-1)}), Y)$ is approximately valid[3]. If we can assume graph structures that represent the relationships of sample points, we can utilize them to define $\mathcal{G}_{\text{KTA}}^{(t)}$. For example, we used the linear combinations of various powers of the adjacency matrix in our experiments.

**Fine Tuning**  After the training using boosting algorithms, we can optionally fine-tune the whole model. For example, if each component of the model is differentiable, we can train it in an end-to-end manner using backpropagation. Because fine-tuning does not increase the Rademacher complexity, it does not worsen the generalization gap bound. Therefore, if fine-tuning does not increase the training error, it does not worsen the test error theoretically. However, because it is not true in all cases, we should compare the model with and without fine-tuning and select the better of the two in practice.

**Computational Complexity**  The memory-efficiency is an advantage of the boosting algorithm. When we train the model without fine-tuning, we do not have to retain intermediate weights and outputs. Therefore, its memory usage is constant w.r.t. $T$, assuming that the memory usage of transformation functions $b^{(t)}$ are the same. This is in contrast to the ordinal GNN models trained in an end-to-end manner. Fine-tuned models use memory proportional to the depth $T$.

## 7  Experiments

To confirm that boosting algorithms can train multi-scale GNNs practically and our theoretical observations reflect practical behaviors, we applied our models, coined *Gradient Boosting Graph*

Table 1: Accuracy of node classification tasks. Numbers denote the $(\text{mean})\pm(\text{standard deviation})(\%)$ of ten runs. $(*)$ All runs failed due to GPU memory errors. $(**)$ We have cited the result of [36]. See Section F.2 for more comprehensive comparisons with other GNNs.

| Model | | Cora | CiteSeer | PubMed |
|---|---|---|---|---|
| GB-GNN | Adj. | $79.9 \pm 0.8$ | $70.5 \pm 0.8$ | $79.4 \pm 0.2$ |
| | Adj. + Fine Tuning | $80.4 \pm 0.8$ | $70.8 \pm 0.8$ | $79.0 \pm 0.5$ |
| | KTA | $80.9 \pm 0.9$ | $73.1 \pm 1.1$ | $79.1 \pm 0.4$ |
| | KTA + Fine Tuning | $82.3 \pm 1.1$ | $70.8 \pm 1.0$ | N.A.$^{(*)}$ |
| GCN$^{(**)}$ | – | $81.5$ | $70.3$ | $79.0$ |

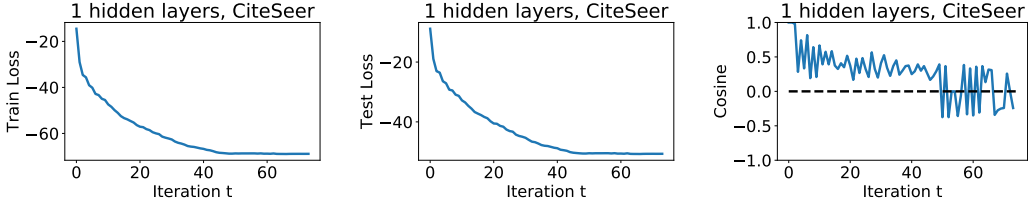

Figure 2: Results of GB-GNN-Adj trained with the CiteSeer dataset. (Left) The transition of the training loss, (Middle) test loss, (Right) angle $\cos\theta^{(t)}$ between weak learners and negative gradients.

*Neural Networks* (GB-GNN), to node classification tasks on citation network. We used Cora [42, 58], CiteSeer [25, 58], and PubMed [58] datasets. We used SAMME [29], an extension of AdaBoost for multi-class classification tasks, as a boosting algorithm. We considered two variants as aggregation functions $\mathcal{G}$: the multiplication model $\mathcal{G}_{\tilde{A}}$ by the augmented normalized adjacency matrix $\tilde{A}$ consisting of a singleton $\mathcal{G}_{\tilde{A}} := \{X \mapsto \tilde{A}X\}$ and the KTA model $\mathcal{G}_{\text{KTA}}$ (we refer to them as GB-GNN-Adj and GB-GNN-KTA, respectively). We employed MLPs with single hidden layers as transformation functions $\mathcal{B}$. See Section E regarding the further experiment setups. In Section F.1, we report the performance of three types of model variants that use (1) MLPs with different layer size, (2) Input Injection, which is another node aggregation strategy similar to GCNII [9], and (3) SAMME.R, a different boosting algorithm.

Table 1 presents the prediction accuracy. It is noteworthy that boosting algorithms greedily train the models and achieve comparable performance to existing GNNs trained in an end-to-end manner by backpropagation. Fine-tuning enhanced the performance of GB-GNN-KTA in the Cora dataset. However, whether it works well depends on model–dataset combinations. One fine-tuned model failed due to the memory error. There are two reasons. First, our implementation naively processes all nodes at once. Second, memory consumption of fine-tuning models increase proportionally to the depth $T$. These problems are not specific to our model but common to end-to-end deep GNN models. We can solve them by node mini-batching.

Figure 2 shows the transition of loss values and angle between the obtained weak learners $f^{(t)}$ and negative gradients $-\nabla\widehat{\mathcal{L}}(\widehat{Y}^{(t)})$ during the training of GB-GNN-Adj using the CiteSeer dataset. Both training and test errors keep decreasing until GB-GNN has grown up to be a deep model with as many as 40 weak learners. Accordingly, the angle is acute within this period, meaning that the w.l.c. is satisfied by Proposition 1. In later iterations, the training and test errors saturate, and the angle fluctuates. These behaviors are consistent with Theorem 2, which implies that the training and test error bounds monotonically decrease under the w.l.c. We observed similar behaviors for MLPs with various layer sizes.

## 8 Discussion

**Satisfiability of Weak Learning Condition**   The key assumption of our theory is the w.l.c. (Definition 1). Although we have observed in Section 7 that the w.l.c. empirically holds, it is a natural question whether we can *provably* ensure it. To obtain the guaranteed w.l.c., the model must be

sufficiently expressive so that it can approximate all possible values of the negative gradient. We can show that gradient descent can find a weak learner made of an overparameterized MLP that the w.l.c. holds with high probability, by leveraging the optimization analysis in the NTK regime [3, 16] (see Section C for details). However, the guaranteed w.l.c. comes at the cost of large model complexity, as evident the following proposition (see Section A.7 for the proof).

**Proposition 5.** *Let $\mathcal{V}, \mathcal{V}_g \subset \mathbb{R}^N$, and $\alpha > \beta \geq 0$ such that $\{-1, 0, 1\}^N \subset \mathcal{V}_g$. If for any $\boldsymbol{g} \in \mathcal{V}_g$, there exists $Z \in \mathcal{V}$ such that $Z$ satisfies $(\alpha, \beta, \boldsymbol{g})$-w.l.c., then, we have $\mathfrak{R}(\mathcal{V}) \geq \frac{\alpha^2 - \beta^2}{\alpha}$.*

Let $\mathcal{V}_g$ be the set of possible values that can be taken by the negative gradient and $\mathcal{V}$ be the space of outputs of weak learners at a specific iteration. Then, if we want weak learners to satisfy the w.l.c., its Rademacher complexity is inevitably as large as $\Omega(1)$ (assuming that $\alpha$ and $\beta$ are independent of $M$). We leave the problem for future work whether there exists a setting that simultaneously satisfies the following conditions: (1) the w.l.c. (or similar conditions) provably holds, (2) training of weak learners is tractable, and (3) the model has a small complexity (such as the Rademacher complexity).

**Choice of Transformation Functions**   In Section 5, we used an $L$-layered MLP with $L_1$ norm constraints as a transformation function $\mathcal{B}^{(t)}$. The test error bound in Theorem 2 can exponentially depend on $L$ via the constant $D^{(t)}$. Since this problem occurs in inductive MLPs, too, many studies derived generalization bounds that avoid this exponential dependency [4, 44, 66]. We can incorporate them to obtain tighter bounds using Proposition 4.

**Choice of Node Aggregation Functions**   We considered a *linear* aggregation model as $\mathcal{G}^{(t)}$ in Section 7 because GNNs that consist of linear node aggregations and non-linear MLPs is practically popular in the GNN research, such as SGC [67], gfNN [47], and APPNP [37]. Theoretically, [47, 48] claimed that non-linearity between aggregations is not essential for predictive performance.

We can alternatively use *non-linear* aggregation models. For example, consider the model $X \mapsto \sigma(\tilde{P}^{(t)} X W)$ with the same $L_1$ constraints as Equation (3) as $\mathcal{G}^{(t)}$, where $\sigma : \mathbb{R} \to \mathbb{R}$ is the ReLU function. Adding the non-linearity $\sigma$ changes two things. First, $\|P^{(t)} X\|_{\mathrm{F}}$ in the bound of Theorem 2 is replaced with $\prod_{s=2}^{t} \|\tilde{P}^{(s)}\|_{\mathrm{op}} \|X\|_{\mathrm{F}}$. It makes the interpretation of the trade-off discussed in Section 5 impossible, and the bound looser, essentially because the bound loses the information of eigenvectors. Second, the bound for Rademacher complexity of $\mathcal{F}^{(t)}$ is multiplied by $2^t$. It changes the condition for the monotonically decreasing test error bound with respect to $T$ from $\alpha_t^{-1} D^{(t)} \|P^{(t)}\|_{\mathrm{op}} = O(\tilde{\varepsilon}^t)$ to a stricter one $\alpha_t^{-1} 2^t D^{(t)} \prod_{s=2}^{t} \|\tilde{P}^{(s)}\|_{\mathrm{op}} = O(\tilde{\varepsilon}^t)$.

With that being said, we do not have a definitive answer whether linear aggregation models are truly superior to non-linear ones — we may be able to use techniques similar to [48] for the first problem and refined analyses could eliminate the $2^t$ term for the second problem.

# 9   Conclusion

In this study, we analyzed a certain type of transductive learning models and derived their optimization and generalization guarantees under the weak learnability condition (w.l.c.). Our idea was to interpret multi-scale GNNs as an ensemble of weak learners and apply boosting theory. As a special case, we showed that a particular type of multi-scale GNNs has a generalization bound that is decreasing with respect to the number of node aggregations under the condition. To the best of our knowledge, this is the first result that multi-scale GNNs provably avoid the over-smoothing from the viewpoint of learning theory. We confirmed that our models, coined GB-GNNs, worked comparably to existing GNNs, and that their empirical behaviors were consistent with theoretical observations. We believe that exploring deeper relationships between the w.l.c. and the underlying graph structures such as graph spectra is a promising direction for future research.

## Broader Impact

**Benefits**   Deepening the theoretical understanding of machine learning models motivates people to explore their advanced usage. For example, the universality of MLPs boosted their usage as a building block of more advanced models as a function approximator (e.g., deep reinforcement

learning). In this study, we investigated the probable optimization and generalization guarantees of GNNs, especially multi-scale GNNs. We expect that this research to broaden the applicability of GNNs in both theoretical and practical situations.

**Potential Risks and Associated Mitigations**   The theoretical understanding of GNNs could result in their misuse, either intentionally or unintentionally. For example, if a GNN is used in social networks, the contamination of biased information on the networks such as fake news could affect the prediction of GNNs, thereby resulting in the promotion of social disruption or unfair treatment to minority groups. The methods devised in the study of fairness could mitigate such misuse of GNNs.

## Acknowledgments and Disclosure of Funding

We would like to thank the following people for insightful discussions: Kohei Hayashi, Masaaki Imaizumi, Masanori Koyama, Takanori Maehara, Kentaro Minami, Atsushi Nitanda, Akiyoshi Sannai, Sho Sonoda, and Yuuki Takai. TS was partially supported by JSPS KAKENHI (18K19793, 18H03201, and 20H00576), Japan Digital Design, and JST CREST.

## Footnotes

[1]As usual, we can take into account of bias terms by preprocessing the input as $\mathbb{R}^C \ni \boldsymbol{x} \mapsto (\boldsymbol{x}, 1) \in \mathbb{R}^{C+1}$

[2]Let $D$ be the degree matrix of $G$ and $\tilde{D} = D + I$. We define $\tilde{A} := \tilde{D}^{-\frac{1}{2}}(A + I)\tilde{D}^{-\frac{1}{2}}$ [36].

[3]When the task is a classification in which $\mathcal{Y} = [K]$, we identify $Y \in \mathcal{Y}^N$ with the matrix consisting of one-hot vectors: $\tilde{Y} = (\tilde{y}_1, \ldots, \tilde{y}_N)^\top \in \mathbb{R}^{N \times K}$ with $\tilde{y}_{nk} = \mathbf{1}\{y_n = k\}$.

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
