[Supplementary Material]

# Appendices

This is the supplemental material for *Optimization and Generalization Analysis of Transduction through Gradient Boosting and Application to Multi-scale Graph Neural Networks*.

## A   Proof of Theorems and Propositions

We give proofs for the theorems and propositions in the order they appeared in the main paper.

### A.1   Proof of Proposition 1

We prove the more detailed claim. Proposition 1 is a part of the following proposition. The third condition below means that the prediction $Z$ is better than a random guess as a solution to the binary classification problem on the training dataset weighed by $w_n$'s. The proof for the equivalence of the second and third conditions are similar to that of [10, Theorem 1]

**Proposition 6.** *Let $Z, \boldsymbol{g} \in \mathbb{R}^N$ such that $\boldsymbol{g} \neq 0$. The followings are equivalent*

1. *There exist $\alpha, \beta$ such that $\alpha > \beta \geq 0$ and $Z$ satisfies $(\alpha, \beta, \boldsymbol{g})$-w.l.c.*

2. *$\langle Z, \boldsymbol{g} \rangle > 0$.*

*Under the condition, for any $r \in [\sin^2 \theta, 1)$, we can take $\alpha := C$, $\beta := rC$, and $\|Z - \alpha \boldsymbol{g}\|_2 = \beta \|\boldsymbol{g}\|_2$, where*

$$\cos \theta = \frac{\langle Z, \boldsymbol{g} \rangle}{\|Z\|_2 \|\boldsymbol{g}\|_2}, \quad C := \frac{\langle Z, \boldsymbol{g} \rangle \pm \sqrt{\langle Z, \boldsymbol{g} \rangle^2 - (1 - r^2)\|Z\|^2 \|\boldsymbol{g}\|^2}}{(1 - r^2)\|\boldsymbol{g}\|_2^2}.$$

*Suppose further $Z \in \{\pm 1\}^N$, then, the conditions 1 and 2 are equivalent to*

3. *There exists $\delta \in (0, 1]$ such that $\sum_{n=1}^N w_n \mathbf{1}\{\text{sign}(\boldsymbol{g}_n) \neq Z_n\} \leq \frac{1-\delta}{2}$ where $w_n = \frac{\boldsymbol{g}_n}{\|\boldsymbol{g}\|_1}$.*

*Proof.* ($1. \implies 2.$) We have

$$\|Z - \alpha \boldsymbol{g}\|_2 \leq \beta \|\boldsymbol{g}\|_2 \iff \|Z\|_2^2 - 2\alpha \langle Z, \boldsymbol{g} \rangle + \alpha^2 \|\boldsymbol{g}\|_2^2 \leq \beta^2 \|\boldsymbol{g}\|_2^2$$

$$\iff \langle Z, \boldsymbol{g} \rangle \geq \frac{\|Z\|_2^2}{2\alpha} + \frac{\alpha^2 - \beta^2}{2\alpha} \|\boldsymbol{g}\|_2^2 > 0. \tag{5}$$

($2. \implies 1.$) For $k > 0$, we define

$$\tilde{r}(k) := \frac{\|Z - k\boldsymbol{g}\|_2}{k\|\boldsymbol{g}\|_2}.$$

Then, by direct computation, we have

$$\tilde{r}(k)^2 = \frac{\|Z\|_2^2}{\|\boldsymbol{g}\|_2^2} \left( \gamma - \frac{\langle Z, \boldsymbol{g} \rangle}{\|Z\|_2^2} \right)^2 + 1 - \frac{\langle Z, \boldsymbol{g} \rangle^2}{\|Z\|_2^2 \|\boldsymbol{g}\|_2^2}.$$

where $\gamma := k^{-1}$. Therefore, $\tilde{r}(k)$ is a quadratic function of $\gamma$ that takes the minimum value $\sin^2 \theta$ at $\gamma = \frac{\langle Z, \boldsymbol{g} \rangle}{\|Z\|_2^2} > 0$. Therefore, for any $r \in [\sin^2 \theta, 1)$ there exists $k_0 > 0$ such that $\tilde{r}(k_0) = r$. Then, by setting $\alpha := k_0$ and $\beta := rk_0$, we have $\alpha > \beta \geq 0$ and

$$\|Z - \alpha \boldsymbol{g}\|_2 = \tilde{r}(k_0)\alpha\|\boldsymbol{g}\|_2 = r\alpha\|\boldsymbol{g}\|_2 = \beta\|\boldsymbol{g}\|_2.$$

By solving $\tilde{r}(k_0) = r$, we obtain $\alpha = C$ and $\beta = rC$.

($1. \implies 3.$) Define $w^+ := \sum_{n=1}^N w_n \mathbf{1}\{Z_n = \text{sign}(\boldsymbol{g}_n)\}$ and $w^- := \sum_{n=1}^N w_n \mathbf{1}\{Z_n \neq \text{sign}(\boldsymbol{g}_n)\}$. By definition, we have $w^+ + w^- = \|w\|_1 = 1$.

$$\langle Z, \boldsymbol{g} \rangle = \sum_{n=1}^N Z_n \boldsymbol{g}_n$$

$$= \sum_{n=1}^{N} Z_n w_n \|\boldsymbol{g}\|_1 \operatorname{sign}(\boldsymbol{g}_n)$$
$$= \|\boldsymbol{g}\|_1 (w^+ - w^-). \tag{6}$$

Therefore, using the reformulation Equation (5) of the assumption, we have

$$
\begin{aligned}
w^+ - w^- &= \frac{\langle Z, \boldsymbol{g} \rangle}{\|\boldsymbol{g}\|_1} \\
&\geq \frac{\langle Z, \boldsymbol{g} \rangle}{\|\boldsymbol{g}\|_2 \sqrt{N}} \quad (\because \text{Cauchy–Schwraz inequality)}) \\
&\geq \frac{\sqrt{N}}{2\alpha \|\boldsymbol{g}\|_2} + \frac{\alpha^2 - \beta^2}{2\alpha} \frac{\|\boldsymbol{g}\|_2}{\sqrt{N}} \quad (\because \text{Equation (5) and } \|Z\|_2^2 = N) \\
&\geq 2\sqrt{\frac{1}{2\alpha} \frac{\alpha^2 - \beta^2}{2\alpha}} \quad (\because \text{AM–GM inequality}) \\
&= \sqrt{1 - \frac{\beta^2}{\alpha^2}}.
\end{aligned}
$$

Set $\delta := \sqrt{1 - \frac{\beta^2}{\alpha^2}}$. By the assumption $\alpha > \beta \geq 0$, we have $\delta \in (0, 1]$. Therefore, we have

$$
\begin{aligned}
w^+ &= \frac{1}{2}(w^+ + w^-) + \frac{1}{2}(w^+ - w^-) \\
&\geq \frac{1}{2}(w^+ + w^-) + \frac{\delta}{2} \\
&= \frac{1 + \delta}{2},
\end{aligned}
$$

which is equivalent to $w^- \leq \frac{1}{2}(1 - \delta)$.

$(3. \implies 2.)$   Using the same argument as Equation (6), we have

$$\langle Z, \boldsymbol{g} \rangle = \|\boldsymbol{g}\|_1 (w^+ - w^-).$$

By the assumption, we have

$$
\begin{aligned}
w^- &\leq \frac{1 - \delta}{2}(w^+ + w^-) \\
\iff w^+ &\geq \frac{1 + \delta}{2}(w^+ + w^-) \\
\iff w^+ - \frac{w^+ + w^-}{2} &\geq \frac{1 + \delta}{2}(w^+ + w^-) - \frac{w^+ + w^-}{2} \\
\iff \frac{1}{2}(w^+ - w^-) &\geq \frac{\delta}{2}(w^+ + w^-)
\end{aligned}
$$

Therefore, we have

$$
\begin{aligned}
\langle Z, \boldsymbol{g} \rangle &= \|\boldsymbol{g}\|_1 (w^+ - w^-) \\
&= \|\boldsymbol{g}\|_1 \delta (w^+ + w^-) \\
&= \|\boldsymbol{g}\|_1 \delta > 0.
\end{aligned}
$$

$\square$

## A.2   Proof of Theorem 1

**Assumption 1.** $\ell : \widehat{\mathcal{Y}} \times \mathcal{Y} \to \mathbb{R}$ is a non-negative $C^2$ convex funxtion with respect to the first variable and satisfies $|\nabla_{\hat{y}}^2 \ell(\hat{y}, y)| \leq A$ for all $\hat{y} \in \widehat{\mathcal{Y}}$ and $y \in \mathcal{Y}$.

**Proposition 7.** *The sigmoid cross entropy loss $\ell_\sigma$ satisfies Assumption 1 with $A = \frac{1}{4}$.*

**Lemma 1.** *Suppose the loss function $\ell$ satisfies Assumption 1 with $A > 0$. Define the training error $\widehat{\mathcal{L}}$ by $\widehat{\mathcal{L}}(\widehat{Y}) := \frac{1}{M}\sum_{n=1}^{M}\ell(\hat{y}_n, y_n)$ for $\widehat{Y}^\top = (\hat{y}_1, \ldots, \hat{y}_N)^\top$. Suppose Algorithm 1 with the learning rate $\eta^{(t)} = \frac{1}{A\alpha_t}$ finds a weak learner $f^{(t)}$ for any $t \in [T]$. Then, we have*

$$\sum_{t=1}^{T}\gamma_t\|\nabla\widehat{\mathcal{L}}(\widehat{Y}^{(t)})\|_{\mathrm{F}}^2 \le \frac{2A\widehat{\mathcal{L}}(\widehat{Y}^{(1)})}{M}.$$

*Proof.* First, we define $C_f^{(t)} := (2\alpha_t)^{-1}$ and $C_{\mathcal{L}}^{(t)} := \frac{\alpha_t^2 - \beta_t^2}{2\alpha_t}$. Note that we have by definition

$$\gamma_t = 4C_f^{(t)}C_{\mathcal{L}}^{(t)}. \tag{7}$$

We denote $Z^{(t)\top} = (z_1^{(t)}, \ldots, z_N^{(t)})^\top := \frac{1}{M}f^{(t)}(X)^\top$ and $(\hat{y}_1^{(t)}, \ldots, \hat{y}_N^{(t)})^\top := \widehat{Y}^{(t)\top}$. Since $Z^{(t)}$ satisfies $(\alpha_t, \beta_t, -\nabla\widehat{\mathcal{L}}(\widehat{Y}^{(t-1)}))$-w.l.c., we have

$$\|Z^{(t)} + \alpha_t\nabla\widehat{\mathcal{L}}(\widehat{Y}^{(t-1)})\|_{\mathrm{F}} \le \beta_t\|\nabla\widehat{\mathcal{L}}(\widehat{Y}^{(t-1)})\|_{\mathrm{F}}$$

$$\Longleftrightarrow \|Z^{(t)}\|_{\mathrm{F}}^2 + 2\alpha_t\langle Z^{(t)}, \nabla\widehat{\mathcal{L}}(\widehat{Y}^{(t-1)})\rangle + \alpha_t^2\|\nabla\widehat{\mathcal{L}}(\widehat{Y}^{(t-1)})\|_{\mathrm{F}}^2 \le \beta_t^2\|\nabla\widehat{\mathcal{L}}(\widehat{Y}^{(t-1)})\|_{\mathrm{F}}^2$$

$$\Longleftrightarrow \langle Z^{(t)}, \nabla\widehat{\mathcal{L}}(\widehat{Y}^{(t-1)})\rangle + C_f^{(t)}\|Z^{(t)}\|_{\mathrm{F}}^2 \le -C_{\mathcal{L}}^{(t)}\|\nabla\widehat{\mathcal{L}}(\widehat{Y}^{(t-1)})\|_{\mathrm{F}}^2. \tag{8}$$

Since $\eta^{(t)} = \frac{2C_f^{(t)}}{A}$, we have

$$\langle\nabla\widehat{\mathcal{L}}(\widehat{Y}^{(t-1)}), Z^{(t)}\rangle + \frac{A\eta^{(t)}}{2}\|Z^{(t)}\|_{\mathrm{F}}^2 \le -C_{\mathcal{L}}^{(t)}\|\nabla\widehat{\mathcal{L}}(\widehat{Y}^{(t-1)})\|_{\mathrm{F}}^2.$$

By Taylors' theorem, and Assumption 1, we have

$$\ell(\hat{y}_n^{(t)}, y_n) \le \ell(\hat{y}_n^{(t-1)}, y_n) + \langle\nabla_{\hat{y}}\ell(\hat{y}_n^{(t-1)}, y_n), \hat{y}_n^{(t)} - \hat{y}_n^{(t-1)}\rangle + \frac{A}{2}\|\hat{y}_n^{(t)} - \hat{y}_n^{(t-1)}\|_2^2.$$

By taking the average in terms of $n$, we have

$$\widehat{\mathcal{L}}(\widehat{Y}^{(t)}) \le \widehat{\mathcal{L}}(\widehat{Y}^{(t-1)}) + \frac{1}{M}\sum_{n=1}^{M}\langle\nabla_{\hat{y}}\ell(\hat{y}_n^{(t-1)}, y_n), y_n^{(t)} - y_n^{(t-1)}\rangle + \frac{A}{2M}\sum_{n=1}^{M}\|y_n^{(t)} - y_n^{(t-1)}\|_2^2.$$

By the definition of $\widehat{Y}^{(t)}$'s, we have

$$\hat{y}_n^{(t)} - \hat{y}_n^{(t-1)} = \eta^{(t)}Mz_n^{(t)}.$$

Therefore, we have

$$\widehat{\mathcal{L}}(\widehat{Y}^{(t)}) \le \widehat{\mathcal{L}}(\widehat{Y}^{(t-1)}) + \eta^{(t)}\sum_{n=1}^{M}\langle\nabla_{\hat{y}}\ell(\hat{y}_n^{(t-1)}, y_n), z_n^{(t)}\rangle + \frac{1}{2}\eta^{(t)2}AM\sum_{n=1}^{M}\|z_n^{(t)}\|_2^2$$

$$\le \widehat{\mathcal{L}}(\widehat{Y}^{(t-1)}) + \eta^{(t)}\sum_{n=1}^{N}\langle\nabla_{\hat{y}}\ell(\hat{y}_n^{(t-1)}, y_n), z_n^{(t)}\rangle + \frac{1}{2}\eta^{(t)2}AM\sum_{n=1}^{N}\|z_n^{(t)}\|_2^2$$

$$\le \widehat{\mathcal{L}}(\widehat{Y}^{(t-1)}) + \eta^{(t)}M\langle\nabla_{\hat{y}}\widehat{\mathcal{L}}(\widehat{Y}^{(t-1)}), Z^{(t)}\rangle + \frac{1}{2}\eta^{(t)2}AM\|Z^{(t)}\|_{\mathrm{F}}^2$$

$$\le \widehat{\mathcal{L}}(\widehat{Y}^{(t-1)}) - \eta^{(t)}MC_{\mathcal{L}}^{(t)}\|\nabla\widehat{\mathcal{L}}(\widehat{Y}^{(t-1)})\|_{\mathrm{F}}^2 \quad (\because \text{Equation (8)})$$

$$= \widehat{\mathcal{L}}(\widehat{Y}^{(t-1)}) - \frac{2MC_f^{(t)}C_{\mathcal{L}}^{(t)}}{A}\|\nabla\widehat{\mathcal{L}}(\widehat{Y}^{(t-1)})\|_{\mathrm{F}}^2 \quad (\because \text{Definition of } \eta^{(t)})$$

$$= \widehat{\mathcal{L}}(\widehat{Y}^{(t-1)}) - \frac{M\gamma_t}{2A}\|\nabla\widehat{\mathcal{L}}(\widehat{Y}^{(t-1)})\|_{\mathrm{F}}^2 \quad (\because \text{Equation (7)})$$

Rearranging the term, we get

$$\gamma_t\|\nabla\widehat{\mathcal{L}}(\widehat{Y}^{(t-1)})\|_{\mathrm{F}}^2 \le \frac{2A}{M}\left(\widehat{\mathcal{L}}(\widehat{Y}^{(t-1)}) - \widehat{\mathcal{L}}(\widehat{Y}^{(t)})\right).$$

By taking the summation in terms of $t$, we have

$$\sum_{t=1}^{T}\gamma_t\|\nabla\widehat{\mathcal{L}}(\widehat{Y}^{(t)})\|_{\mathrm{F}}^2 \leq \frac{2A}{M}\left(\widehat{\mathcal{L}}(\widehat{Y}^{(1)}) - \widehat{\mathcal{L}}(\widehat{Y}^{(T+1)})\right) \leq \frac{2A\widehat{\mathcal{L}}(\widehat{Y}^{(1)})}{M}.$$

We used the non-negativity of the loss function $\widehat{\mathcal{L}}$ in the final inequality. $\qquad\square$

**Lemma 2.** *Assume the loss function is the cross entropy loss. Let $\delta \geq 0$. For any $\widehat{Y} \in \widehat{\mathcal{Y}}^N$, we have*

$$\widehat{\mathcal{R}}(\widehat{Y}) \leq (1 + e^\delta)\|\nabla\widehat{\mathcal{L}}(\widehat{Y})\|_{2,1}$$

*Proof.* Same as [20, Proposition C]. $\qquad\square$

*Proof of Theorem 1.* Since we use the cross entropy loss, by Lemma 2, we have

$$\widehat{\mathcal{R}}(\widehat{Y}) \leq (1 + e^\delta)\|\nabla\widehat{\mathcal{L}}(\widehat{Y})\|_{2,1} \qquad (9)$$

By the definition of $\widehat{Y}$, $t^*$, and $\Gamma$, we have

$$\begin{aligned}
\Gamma_T\|\nabla\widehat{\mathcal{L}}(\widehat{Y})\|_{2,1} &= \left(\sum_{t=1}^{T}\gamma_t\right)\|\nabla\widehat{\mathcal{L}}(\widehat{Y}^{(t^*)})\|_{2,1} \\
&\leq \sum_{t=1}^{T}\gamma_t\|\nabla\widehat{\mathcal{L}}(y^{(t)})\|_{2,1}^2 \\
&\leq \sum_{t=1}^{T}\gamma_t\|\nabla\widehat{\mathcal{L}}(y^{(t)})\|_{\mathrm{F}}^2 \qquad (10)
\end{aligned}$$

From Lemma 1 with $A = \frac{1}{4}$ (Proposition 7), we have

$$\sum_{t=1}^{T}\gamma_t\|\nabla\widehat{\mathcal{L}}(\widehat{Y}^{(t)})\|_{\mathrm{F}}^2 \leq \frac{\widehat{\mathcal{L}}[\widehat{Y}^{(1)}]}{2M} \qquad (11)$$

Combining Equation (9), Equation (10), and Equation (11), we have

$$\widehat{\mathcal{R}}(\widehat{Y}) \leq \frac{(1 + e^\delta)\widehat{\mathcal{L}}[\widehat{Y}^{(1)}]}{2M\Gamma_T}.$$

$\qquad\square$

## A.3 Proof of Proposition 2

The proof is basically the application of [7, Corollary 1] to our setting. We recall the definition of the transductive Rademacher complexity introduced by [7].

**Definition 2** (Transductive Rademacher Complexity). *For $p \in [0, \frac{1}{2}]$ and $\mathcal{V} \subset \mathbb{R}^N$, we define*

$$\mathfrak{R}(\mathcal{V}, p) := Q\mathbb{E}_{\boldsymbol{\sigma}}\left[\sup_{v \in V} \boldsymbol{\sigma} \cdot v\right],$$

*Here, $Q = \frac{1}{M} + \frac{1}{U}$ and $\boldsymbol{\sigma} = (\sigma_1, \ldots, \sigma_N)$ is an sequence of i.i.d. random variables whose distribution is $\mathbb{P}(\sigma_i = 1) = \mathbb{P}(\sigma_i = -1) = p$ and $\mathbb{P}(\sigma_i = 0) = 1 - 2p$. In particular, we denote $\mathfrak{R}(\mathcal{V}) := \mathfrak{R}(\mathcal{V}, p_0)$ where $p_0 = \frac{MU}{(M+U)^2}$.*

We introduce the notion of the (weighed) sum of sets.

**Definition 3.** *For $\mathcal{V}_1, \ldots, \mathcal{V}_T \subset \mathbb{R}^N$ and $\alpha_1, \ldots, \alpha_T \in \mathbb{R}$, we define their (weighted) sum $\sum_{t=1}^{T}\alpha_t\mathcal{V}_t$ by*

$$\sum_{t=1}^{T}\alpha_t\mathcal{V}_t := \left\{\sum_{t=1}^{T}\alpha_t v_t \,\middle|\, v_t \in \mathcal{V}_t\right\}.$$

We define the hypothesis space $\mathcal{H}$ by $\mathcal{H} := \sum_{t=1}^{T} \eta^{(t)} \mathcal{F}^{(t)}$. Note that any output $\widehat{Y}$ of Algorithm 1 satisfies $\widehat{Y} \in \mathcal{H}$. We can compute the Rademacher complexity of the sum similarly to the inductive case.

**Proposition 8.** *For $\mathcal{V}_1, \mathcal{V}_2 \subset \mathbb{R}^N$, $a_1, a_2 \in \mathbb{R}$ and $p \in [0, \frac{1}{2}]$, we have $\mathfrak{R}(a_1 \mathcal{V}_1 + a_2 \mathcal{V}_2) \leq |a_1| \mathfrak{R}(\mathcal{V}_1) + |a_2| \mathfrak{R}(\mathcal{V}_2)$.*

*Proof.* Take any realization $\boldsymbol{\sigma}$ of the $N$ i.i.d. transductive Rademacher variable of parameter $p$. For any $v = a_1 v_1 + a_2 v_2 \in a_1 \mathcal{V}_1 + a_2 \mathcal{V}_2$ ($v_1 \in \mathcal{V}_1$, $v_2 \in \mathcal{V}_2$), we have

$$\langle \boldsymbol{\sigma}, v \rangle = \langle \boldsymbol{\sigma}, a_1 v_1 \rangle + \langle \boldsymbol{\sigma}, a_2 v_2 \rangle \leq |a_1| \sup_{v_1 \in \mathcal{V}_1} \langle \boldsymbol{\sigma}, v_1 \rangle + |a_2| \sup_{v_1 \in \mathcal{V}_2} \langle \boldsymbol{\sigma}, v_2 \rangle.$$

By taking the supremum of $v$, we have

$$\sup_{v \in \mathcal{V}_1 + \mathcal{V}_2} \langle \boldsymbol{\sigma}, v \rangle \leq |a_1| \sup_{v_1 \in \mathcal{V}_1} \langle \boldsymbol{\sigma}, v_1 \rangle + |a_2| \sup_{v_2 \in \mathcal{V}_2} \langle \boldsymbol{\sigma}, v_2 \rangle.$$

The proposition follows by taking the expectation with respect to $\boldsymbol{\sigma}$. $\qquad\square$

*Proof of Proposition 2.* Let $\mathcal{H} = \sum_{t=1}^{T} \eta^{(t)} \mathcal{F}^{(t)}$. By [7, Corollary 1], with probability of at least $1 - \delta'$ for all $\widehat{Y}' \in \mathcal{H}$, we have

$$\mathcal{R}(\widehat{Y}') \leq \widehat{\mathcal{R}}(\widehat{Y}') + \mathfrak{R}(\mathcal{H}) + c_0 Q \sqrt{M \wedge U} + \sqrt{\frac{SQ}{2} \log \frac{1}{\delta'}}.$$

Since the output $\widehat{Y}$ of Algorithm 1 satisfies $\widehat{Y} \in \mathcal{H}$, we have

$$\mathcal{R}(\widehat{Y}) \leq \widehat{\mathcal{R}}(\widehat{Y}) + \mathfrak{R}(\mathcal{H}) + c_0 Q \sqrt{M \wedge U} + \sqrt{\frac{SQ}{2} \log \frac{1}{\delta'}}. \tag{12}$$

By Proposition 8, we have

$$\mathfrak{R}(\mathcal{H}) \leq \sum_{t=1}^{T} \eta^{(t)} \mathfrak{R}(\mathcal{F}^{(t)}). \tag{13}$$

Combining Equation (12) and Equation (13) concludes the proof. $\qquad\square$

### A.4 Proof of Proposition 3

We shall prove Proposition 9, which is more general than Proposition 3. To formulate it, we first introduce the variant of the transductive Rademacher complexity.

**Definition 4** ((Symmetrized) Transductive Rademacher Complexity). *For $\mathcal{V} \subset \mathbb{R}^N$ and $p \in [0, 1/2]$, we define the symmetrized transductive Rademacher complexity $\overline{\mathfrak{R}}(\mathcal{V}, p)$ by*

$$\overline{\mathfrak{R}}(\mathcal{V}, p) := Q \mathbb{E}_{\boldsymbol{\sigma}} \left[ \sup_{v \in \mathcal{V}} |\langle \boldsymbol{\sigma}, v \rangle| \right].$$

*We denote $\overline{\mathfrak{R}}(\mathcal{V}) := \overline{\mathfrak{R}}(\mathcal{V}, p_0)$ for $p_0 = \frac{MU}{(M+U)^2}$. For $\mathcal{F} \subset \{\mathcal{X} \to \widehat{\mathcal{Y}}\}$, we denote $\overline{\mathfrak{R}}(\mathcal{F}, p) := \overline{\mathfrak{R}}(\mathcal{V}, p)$ where $\mathcal{V} = \{(f(X_1), \ldots, f(X_N)))^{\top} \mid f \in \mathcal{F}\}$, where $X_1, \ldots X_N$ are feature vectors of the given training dataset defined in Section 4.1.*

We refer to the transductive Rademacher complexity defined in Definition 2 as the *unsymmetrized* transductive Rademacher complexity if necessary[1]. Note that we have by definition

$$\mathfrak{R}(\mathcal{V}, p) \leq \overline{\mathfrak{R}}(\mathcal{V}, p). \tag{14}$$

Using the concept of the symmetrized transductive Rademacher complexity, we state the main proposition of this section.

**Proposition 9.** *Let $p \in [0, 1/2]$. Suppose we use $\mathcal{G}^{(t)}$ and $\mathcal{B}^{(t)}$ defined by Equation (1) and Equation (2) as a model. Define $D^{(t)} = 2\sqrt{2}(2\tilde{B}^{(t)})^{L-1}\prod_{s=2}^{t} \tilde{C}^{(s)}$ and $P^{(t)} := \prod_{s=2}^{t} \tilde{P}^{(s)}$. Then, we have*

$$\overline{\mathfrak{R}}(\mathcal{F}^{(t)}, p) \leq \sqrt{2p}QB^{(t)}\|P^{(t)}X\|_{\mathrm{F}}.$$

We shall prove this proposition in the end of this section. Reference [7] proved the contraction property of the unnsymmetrized Rademacher complexity. We prove the contraction property for the symmetrized variant.

**Proposition 10.** *Let $\mathcal{V} \subset \mathbb{R}^N$, $p \in [0, 1/2]$. Suppose $\rho : \mathbb{R} \to \mathbb{R}$ is $L_\rho$-Lipschitz and $\rho(0) = 0$. Then, we have*

$$\overline{\mathfrak{R}}(\rho \circ \mathcal{V}, p) \leq 2L_\rho\overline{\mathfrak{R}}(\mathcal{V}, p),$$

*where $\rho \circ \mathcal{V} := \{(\rho(v_1), \ldots, \rho(v_N))^\top \mid \boldsymbol{v} = (v_1, \ldots, v_N)^\top \in \mathcal{V}\}$.*

*Proof.* First, by the definition $\overline{\mathfrak{R}}$ and $\rho(0) = 0$, we have

$$\overline{\mathfrak{R}}(\mathcal{V} \cup \{0\}, p) = \overline{\mathfrak{R}}(\mathcal{V}, p),$$
$$\overline{\mathfrak{R}}(\rho \circ (\mathcal{V} \cup \{0\}), p) = \overline{\mathfrak{R}}((\rho \circ \mathcal{V}) \cup \{0\}, p) = \overline{\mathfrak{R}}(\rho \circ \mathcal{V}, p).$$

Therefore, we can assume without loss of generality that $0 \in \mathcal{V}$. Then, we have

$$\begin{aligned}
\overline{\mathfrak{R}}(\rho \circ \mathcal{V}, p) &= Q\mathbb{E}_{\boldsymbol{\sigma}} \sup_{v \in \mathcal{V}} |\langle \boldsymbol{\sigma}, \rho(v) \rangle| \\
&\leq Q\mathbb{E}_{\boldsymbol{\sigma}} \sup_{v \in \mathcal{V}} \langle \boldsymbol{\sigma}, \rho(v) \rangle + Q\mathbb{E}_{\boldsymbol{\sigma}} \sup_{v \in \mathcal{V}} \langle \boldsymbol{\sigma}, -\rho(v) \rangle \\
&= \mathfrak{R}(\rho \circ \mathcal{V}, p) + \mathfrak{R}(-\rho \circ \mathcal{V}, p),
\end{aligned} \tag{15}$$

where $\boldsymbol{\sigma} = (\sigma_1, \ldots, \sigma_N)$ are the i.i.d. transducive Rademacher variables of parameter $p$. We used $0 \in \mathcal{V}$ and $\rho(0) = 0$ in the inequality above. By the contraction property of the unsymmetrized transductive Rademacher complexity ([7, Lemma 1]), we have

$$\mathfrak{R}(\rho \circ \mathcal{V}, p) \leq L_\rho\mathfrak{R}(\mathcal{V}, p)$$
$$\mathfrak{R}(-\rho \circ \mathcal{V}, p) \leq L_\rho\mathfrak{R}(\mathcal{V}, p)$$

By combining them with Equation (14) and Equation (15), we have

$$\overline{\mathfrak{R}}(\rho \circ \mathcal{V}, p) \leq 2L_\rho\mathfrak{R}(\mathcal{V}, p) \leq 2L_\rho\overline{\mathfrak{R}}(\mathcal{V}, p).$$

$\square$

*Proof of Proposition 9.* The proof is the extension of [18, Exercises 3.11] to the transductive and multi-layer setting. See also the proof of [20, Theorem 3]. First, we note that the multiplication $X \mapsto \tilde{P}^{(s)}X$ in $\mathcal{G}^{(s)}$ and the multiplication $X \mapsto XW^{(s-1)} \in \mathbb{R}^{C \times C}$ in $\mathcal{G}^{(s-1)}$ are commutative operations. Therefore, we have

$$\mathcal{F}^{(t)}(X) := \{(\boldsymbol{z}_1, \ldots, \boldsymbol{z}_N) \mid f \in \mathcal{B}_{\mathrm{base}}^{(t)}, \|W_{\cdot c}^{(s)}\|_1 \leq C^{(s)} \text{ for all } c \in [C] \text{ and } s = 2, \ldots, t\},$$

where $\boldsymbol{z}_n := f(\boldsymbol{x}_n W^{(2)} \cdots W^{(t)})$ and $\boldsymbol{x_n} := (P^{(t)}X)_n \in \mathbb{R}^C$. Therefore, it is sufficient that we first prove the proposition by assuming $\tilde{P}^{(s)} = I_N$ for all $s = 2, \ldots, t$ and then replace $X$ with $P^{(t)}X$.

We define $\mathcal{J}^{(s)} \subset \mathbb{R}^N$ be the set of possible values of any channel of the $s$-th representations and $\mathcal{H}^{(l)} \subset \mathbb{R}^N$ be the set of possible values of any output channel of the $l$-th layer of an MLP. More concretely, we define

$$\mathcal{J}^{(1)} := \{X_{\cdot c} \mid c \in [C]\},$$
$$\mathcal{J}^{(s+1)} := \left\{ \sum_{c=1}^{C} \boldsymbol{z}_c w_c \ \middle| \ \boldsymbol{z}_c \in \mathcal{J}^{(s)}, \|w\|_1 \leq \tilde{C}^{(s+1)} \right\},$$

for $s = 1, \ldots t - 1$. Similarly, we define

$$\mathcal{H}^{(1)} := \mathcal{J}^{(t)},$$

$$\tilde{\mathcal{H}}^{(l+1)} := \left\{ \sum_{c=1}^{C_{l+1}} \boldsymbol{z}_c w_c \ \middle| \ \boldsymbol{z}_c \in \mathcal{H}^{(l)}, \|w\|_1 \leq \tilde{B}^{(t)} \right\},$$

$$\mathcal{H}^{(l+1)} := \sigma \circ \tilde{\mathcal{H}}^{(l+1)} = \{\sigma(\boldsymbol{z}) \mid \boldsymbol{z} \in \tilde{\mathcal{H}}^{(l+1)}\}.$$

for $l = 1, \ldots, L$. By the definition of $\mathcal{F}^{(t)}$, we have $\{f(X) \mid f \in \mathcal{F}^{(t)}\} = \tilde{\mathcal{H}}^{(L+1)}$. On one hand, we can bound the Rademacher complexity of $\tilde{\mathcal{H}}^{(l)}$ as

$$Q^{-1}\overline{\mathfrak{R}}(\tilde{\mathcal{H}}^{(l+1)}, p) = \mathbb{E}_{\boldsymbol{\sigma}} \left[ \sup_{\|w\|_1 \leq \tilde{B}^{(l)}, Z_{\cdot c} \in \mathcal{H}^{(l)}} \left| \sum_{n=1}^{N} \sigma_n \sum_{c=1}^{C_{l+1}} Z_{nc} w_c \right| \right]$$

$$= \mathbb{E}_{\boldsymbol{\sigma}} \left[ \sup_{\|w\|_1 \leq \tilde{B}^{(l)}, Z_{\cdot c} \in \mathcal{H}^{(l)}} \left| \sum_{c=1}^{C_{l+1}} w_c \sum_{n=1}^{N} \sigma_n Z_{nc} \right| \right]$$

$$= \tilde{B}^{(t)} \mathbb{E}_{\boldsymbol{\sigma}} \left[ \sup_{Z \in \mathcal{H}^{(l)}} \left| \sum_{n=1}^{N} \sigma_n Z_n \right| \right]$$

$$= \tilde{B}^{(t)} Q^{-1} \overline{\mathfrak{R}}(\mathcal{H}^{(l)}, p). \tag{16}$$

On the other hand, since $\sigma$ is 1-Lipschitz, by the contraction property (Proposition 10), we bound the Rademacher complexity of $\mathcal{H}^{(l+1)}$ as

$$\overline{\mathfrak{R}}(\mathcal{H}^{(l+1)}, p) \leq 2\overline{\mathfrak{R}}(\tilde{\mathcal{H}}^{(l+1)}, p). \tag{17}$$

By combining Equation (16) and Equation (17), we have the inductive relationship.

$$\overline{\mathfrak{R}}(\mathcal{H}^{(l+1)}, p) \leq 2\tilde{B}^{(t)}\overline{\mathfrak{R}}(\mathcal{H}^{(l)}, p). \tag{18}$$

Using the similar argument to $\mathcal{J}^{(s)}$'s, for $s \in 2, \ldots, t-1$, we have

$$\overline{\mathfrak{R}}(\mathcal{J}^{(s+1)}, p) \leq \tilde{C}^{(s)}\overline{\mathfrak{R}}(\mathcal{J}^{(s)}, p). \tag{19}$$

Let $P_c \in \mathbb{R}^C$ be the projection matrix onto the $c$-th coordinate. Then, for the base step, we can evaluate the Rademacher complexity of $\mathcal{J}^{(1)}$ as

$$Q^{-1}\overline{\mathfrak{R}}(\mathcal{J}^{(1)}, p) = \mathbb{E}_{\boldsymbol{\sigma}} \left[ \max_{c \in [C]} \left| \sum_{n=1}^{N} \sigma_n X_{nc} \right| \right]$$

$$= \mathbb{E}_{\boldsymbol{\sigma}} \left[ \max_{c \in [C]} \left| \left( \sum_{n=1}^{N} \sigma_n X_n \right) P_c \right| \right]$$

$$\leq \mathbb{E}_{\boldsymbol{\sigma}} \left[ \max_{c \in [C]} \|P_c\|_{\mathrm{op}} \left\| \sum_{n=1}^{N} \sigma_n X_n \right\|_2 \right]$$

$$\leq \mathbb{E}_{\boldsymbol{\sigma}} \left\| \sum_{n=1}^{N} \sigma_n X_n \right\|_2$$

$$\leq \sqrt{ \mathbb{E}_{\boldsymbol{\sigma}} \sum_{c=1}^{C} \left( \sum_{n=1}^{N} \sigma_n X_{nc} \right)^2 } \quad (\because \text{Jensen's inequality})$$

$$= \sqrt{ \mathbb{E}_{\boldsymbol{\sigma}} \sum_{c=1}^{C} \sum_{n,m=1}^{N} \sigma_n \sigma_m X_{nc} X_{mc} }$$

$$= \sqrt{ \sum_{c=1}^{C} \sum_{n=1}^{N} 2p(X_{nc})^2 } \tag{20}$$

$$= \sqrt{2p}\|X\|_{\mathrm{F}}. \tag{21}$$

Here, we used in Equation (20) the equality

$$\mathbb{E}_{\boldsymbol{\sigma}} \sigma_m \sigma_n = 2p\delta_{mn}, \tag{22}$$

which is shown by the independence of transductive Rademacher variables. By using the inequalities we have proved so far, we obtain

$$
\begin{aligned}
\overline{\mathfrak{R}}(\mathcal{F}^{(t)}, p) &= \overline{\mathfrak{R}}(\tilde{\mathcal{H}}^{(L+1)}, p) \\
&= 2\overline{\mathfrak{R}}(\mathcal{H}^{(L)}, p) \quad (\because \text{Equation (17)}) \\
&\le 2(2\tilde{B}^{(t)})^{L-1}\overline{\mathfrak{R}}(\mathcal{H}^{(1)}, p) \quad (\because \text{Equation (18)}) \\
&\le 2(2\tilde{B}^{(t)})^{L-1}\overline{\mathfrak{R}}(\tilde{\mathcal{J}}^{(t)}, p) \quad (\because \text{Definition of } \mathcal{H}^{(1)}) \\
&\le 2(2\tilde{B}^{(t)})^{L-1}\left(\prod_{s=2}^{t}\tilde{C}^{(s)}\right)\overline{\mathfrak{R}}(\tilde{\mathcal{J}}^{(1)}, p) \quad (\because \text{Equation (19)}) \\
&\le \sqrt{p}QD^{(t)}\|X\|_{\mathrm{F}}, \quad (\because \text{Equation (21)})
\end{aligned}
$$

where we used $D^{(t)} = 2\sqrt{2}(2\tilde{B}^{(t)})^{L-1}\prod_{s=2}^{t}\tilde{C}^{(s)}$. Therefore, the proposition is true for $\tilde{P}^{(s)} = I_N$ for all $s = 2, \ldots, t$. As stated in the beginning of the proof, we should replace $X$ with $P^{(t)}X$ in the general case. $\qquad\square$

*Proof of Proposition 3.* By applying Proposition 9 with $p = p_0 = \frac{MU}{(M+U)^2}$ and using Equation (14), we have

$$\mathfrak{R}(\mathcal{F}^{(t)}) \le \overline{\mathfrak{R}}(\mathcal{F}^{(t)}, p_0) \le \sqrt{\frac{MU}{(M+U)^2}}QD^{(t)}\|P^{(t)}X\|_{\mathrm{F}} = \frac{D^{(t)}\|P^{(t)}X\|_{\mathrm{F}}}{\sqrt{MU}}.$$

$\qquad\square$

### A.5 Proof of Theorem 2

*Proof.* By Proposition 2, with probability $1 - \delta'$, we have

$$\mathcal{R}(\widehat{Y}) \le \widehat{\mathcal{R}}(\widehat{Y}) + \sum_{t=1}^{T}\eta^{(t)}\mathfrak{R}(\mathcal{F}^{(t)}) + c_0 Q\sqrt{M \wedge U} + \sqrt{\frac{SQ}{2}\log\frac{1}{\delta'}}. \tag{23}$$

By Theorem 1, we have

$$\widehat{\mathcal{R}}(\widehat{Y}) \le \frac{(1 + e^{\delta})\widehat{\mathcal{L}}(\widehat{Y}^{(1)})}{2M\Gamma_T}. \tag{24}$$

By Proposition 3, we have

$$\mathfrak{R}(\mathcal{F}^{(t)}) \le \frac{D^{(t)}\|P^{(t)}X\|_{\mathrm{F}}}{\sqrt{MU}} \tag{25}$$

By applying Equation (24) and Equation (25) to Equation (23) and substituting the learning rate $\eta^{(t)} = \frac{4}{\alpha_t}$, we obtained Equation (4). In particular, when $\Gamma_T = O(T^{\varepsilon})$, the first term of Equation (4) is $O(T^{-\varepsilon})$, which is asymptotically monotonically decreasing (assuming $\delta$, $\widehat{Y}^{(1)}$, and $M$ is independent of $T$). When $\alpha_t^{-1}B^{(t)}\|P^{(t)}\|_{\mathrm{op}}^t = O(\tilde{\varepsilon}^t)$, the second term of Equation (4) is bounded by

$$
\begin{aligned}
\frac{4}{\sqrt{MU}}\sum_{t=1}^{T}\frac{D^{(t)}\|P^{(t)}X\|_{\mathrm{F}}}{\alpha_t} &\le \frac{4\sqrt{2}\|X\|_{\mathrm{F}}}{\sqrt{MU}}\sum_{t=1}^{T}\frac{B^{(t)}\|P^{(t)}\|_{\mathrm{op}}}{\alpha_t} \\
&\lesssim \frac{\|X\|_{\mathrm{F}}}{\sqrt{MU}}\frac{1}{1 - \tilde{\varepsilon}}.
\end{aligned}
$$

The upper bound is independent of $T$ (assuming that $\|X\|_{\mathrm{F}}$, $M$, and $U$ are independent of $T$). $\qquad\square$

## A.6  Proof of Proposition 4

First, we recall the usual (i.e., inductive) version of the Rademacher complexity. We employ the following definition.

**Definition 5** ((Inductive) Empirical Rademacher Complexity)**.** *For $\mathcal{F}_{\text{base}} \subset \{\mathcal{X} \to \widehat{\mathcal{Y}}\}$ and $Z = (z_1, \ldots, z_N) \in \mathcal{X}^N$, we define the (inductive) empirical Rademacher complexity $\widehat{\mathfrak{R}}_{\text{ind}}(\mathcal{F}_{\text{base}})$ conditioned on $Z$ by*

$$\widehat{\mathfrak{R}}_{\text{ind}}(\mathcal{F}_{\text{base}}; Z) := \frac{1}{N}\mathbb{E}_{\boldsymbol{\varepsilon}}\left[\sup_{f \in \mathcal{F}_{\text{base}}} \sum_{n=1}^{N} \varepsilon_n f(z_n)\right],$$

*where $\boldsymbol{\varepsilon} = (\varepsilon_1, \ldots, \varepsilon_N)$ is the i.i.d. Rademacher variables defined by $\mathbb{P}(\varepsilon_i = 1) = \mathbb{P}(\varepsilon_i = -1) = 1/2$.*

*Proof of Proposition 4.* Similarly to Proposition 9 it is sufficient that we first prove the proposition by assuming $\tilde{P}^{(s)} = I_N$ for all $s = 2, \ldots, t$ and then replace $X$ with $P^{(t)}X$. By definition, the transductive Rademacher variable of parameter $p = 1/2$ equals to the (inductive) Rademacher variable. Therefore, we have

$$Q^{-1}\mathfrak{R}(\mathcal{F}^{(t)}, 1/2) = \mathbb{E}_{\boldsymbol{\sigma}}\left[\sup_{f \in \mathcal{F}^{(t)}} \sum_{n=1}^{N} \sigma_n f(X)_n\right]$$

$$= \mathbb{E}_{\boldsymbol{\sigma}}\left[\sup_{f_{\text{base}} \in \mathcal{B}_{\text{base}}^{(t)}, \|W^{(s)}\|_1 \leq \tilde{C}^{(s)}} \sum_{n=1}^{N} \sigma_n f_{\text{base}}(XW^{(2)} \cdots W^{(t)})\right]$$

$$= N\widehat{\mathfrak{R}}_{\text{ind}}(\mathcal{F}_{\text{base}}^{(t)}; X). \tag{26}$$

Since $p_0 := \frac{MU}{(M+U)^2} < 1/2$, by the monotonicity of the transductive Rademacher complexity (see [7] Lemma 1), we have

$$\mathfrak{R}(\mathcal{F}^{(t)}) = \mathfrak{R}(\mathcal{F}^{(t)}, p_0) < \mathfrak{R}(\mathcal{F}^{(t)}, 1/2). \tag{27}$$

The proposition follows from Equation (26) and Equation (27) as follows

$$\mathfrak{R}(\mathcal{F}^{(t)}) < QN\widehat{\mathfrak{R}}_{\text{ind}}(\mathcal{F}_{\text{base}}^{(t)}; X) = \frac{(1+r)^2}{r}\widehat{\mathfrak{R}}_{\text{ind}}(\mathcal{F}_{\text{base}}^{(t)}; X)$$

$\square$

## A.7  Proof of Proposition 5

*Proof.* We denote $p_0 = \frac{MU}{(M+U)^2}$. Let $\sigma_1, \ldots, \sigma_N$ be the i.i.d. transductive Rademacher variable of parameter $p_0$. Since $\{-1, 0, 1\}^N \subset \mathcal{V}_g$, for any realization of $\boldsymbol{\sigma} = (\sigma_1, \ldots, \sigma_N)$, we have $\boldsymbol{\sigma} \in \mathcal{V}_g$. By the assumption, there exists $Z_{\boldsymbol{\sigma}} \in \mathcal{V}$ such that

$$\|Z_{\boldsymbol{\sigma}} - \alpha\boldsymbol{\sigma}\|_2 \leq \beta\|\boldsymbol{\sigma}\|_2.$$

Set $C_f := (2\alpha)^{-1}$ and $C_{\mathcal{L}} := \frac{\alpha^2-\beta^2}{2\alpha}$. Similarly to the proof of Theorem 1, we have

$$\|Z_{\boldsymbol{\sigma}} - \alpha\boldsymbol{\sigma}\|_2 \leq \beta\|\boldsymbol{\sigma}\|_2$$

$$\iff \|Z_{\boldsymbol{\sigma}}\|_2^2 - 2\alpha\langle Z_{\boldsymbol{\sigma}}, \boldsymbol{\sigma}\rangle + \alpha^2\|\boldsymbol{\sigma}\|_2^2 \leq \beta^2\|\boldsymbol{\sigma}\|_2^2$$

$$\iff C_f\|Z_{\boldsymbol{\sigma}}\|_2^2 + C_{\mathcal{L}}\|\boldsymbol{\sigma}\|_2^2 \leq \langle Z_{\boldsymbol{\sigma}}, \boldsymbol{\sigma}\rangle.$$

Therefore, we have

$$Q^{-1}\mathfrak{R}(\mathcal{V}) = \mathbb{E}_{\boldsymbol{\sigma}}\left[\sup_{Z \in \mathcal{F}}\langle \boldsymbol{\sigma}, Z\rangle\right]$$

$$\geq \mathbb{E}_{\boldsymbol{\sigma}}\left[\langle \boldsymbol{\sigma}, Z_{\boldsymbol{\sigma}}\rangle\right]$$

$$\geq \mathbb{E}_{\boldsymbol{\sigma}}\left[C_{\mathcal{L}}\|\boldsymbol{\sigma}\|_2^2 + C_f\|f_{\boldsymbol{\sigma}}\|_2^2\right]$$

$$\geq \mathbb{E}_{\boldsymbol{\sigma}}\left[C_{\mathcal{L}}\|\boldsymbol{\sigma}\|_2^2\right]$$

$$= C_{\mathcal{L}} \cdot 2Np_0.$$

In the last equality, we used Equation (22). Therefore, we have $\mathfrak{R}(\mathcal{F}) \geq 2C_{\mathcal{L}}Np_0 Q = \frac{\alpha^2-\beta^2}{\alpha}$. $\square$

# B  More Related Work

**Generalization Gap Bounds of GNNs**   Reference [23] derived the upper bound of the VC dimension of GNNs. However, the derivation is specific to their model and does not apply to other GNNs. Reference [5] incorporated the idea of Neural Tangent Kernels [13] and derived a generalization gap by reducing it to a kernel regression problem. However, they considered graph prediction problems, where each sample point itself is represented as a graph drawn from some distribution, while our problem is a node prediction problem. Reference [27] derived the generalization gap bounds for node prediction tasks using the stability argument. However, they only considered a GNN with a single hidden layer. It is not trivial to extend their result to multi-layered and multi-scale GNNs. Similarly to our study, [8] employed the (inductive) Rademacher complexity. However, because they did not discuss the optimization guarantee, we cannot directly derive the test error bounds from their analysis.

# C  Provable Satisfiability of Weak Learning Condition using Over-parameterized Models

In this section, we show that there exists a model that for any w.l.c. parameters $\alpha$ and $\beta$, we can find a weak learner which *probably* satisfies the w.l.c. using the gradient descent algorithm. To ensure the w.l.c., the set of transformation functions $\mathcal{B}$ must be sufficiently large so that it can approximate all possible values of the negative gradient $-\nabla\widehat{\mathcal{L}}$ can take. We can accomplish it by leveraging the universal approximation property of MLPs, similarly to graph isomorphism networks (GIN) [31], but for a different purpose. We adopt the recent studies that proved the global convergence of over-parameterized MLPs trained by a tractable algorithm (e.g., [2, 6]).

Let $R \in \mathbb{N}_+$. For the parameter $\Theta = (\theta_{ri}) \in \mathbb{R}^{R \times N}$, we consider an MLP with a single hidden layer $f_\Theta : \mathbb{R}^C \to \mathbb{R}$ defined by

$$f_\Theta(\boldsymbol{x}) := \frac{1}{\sqrt{R}} \sum_{r=1}^{R} a_r \sigma(\theta_{r\cdot}^\top \boldsymbol{x}),$$

where $a_r \in \{-1, 1\}$ and $\sigma$ is the ReLU activation function: $\sigma(x) := x \vee 0$ (we apply ReLU in an element-wise manner for a vector input). We define the set of transformation functions $\mathcal{B} := \{(f_\Theta, \ldots, f_\Theta) \mid \Theta \in \mathbb{R}^{R \times N}\}$. At the $t$-th iteration, given a gradient $\nabla\widehat{\mathcal{L}}(\widehat{Y}^{(t-1)})$, we initialize the model with $a_r \overset{\text{i.i.d.}}{\sim} \text{Unif}(\{-1, 1\})$ and $\theta_{ri} \overset{\text{i.i.d.}}{\sim} \mathcal{N}(0, I)$ independently and train it with the gradient descent to optimize $\Theta$ by minimizing the mean squared error between the output of the model and the properly normalized negative gradient. Using the result of [6], we obtain the following guarantee.

**Proposition 11.** *Suppose Algorithm 1 finds $g^{(s)} \in \mathcal{G}^{(s)}$ ($s \in [t]$) such that $X^{(t)} = g^{(t)} \circ \cdots \circ g^{(1)}(X) = [\boldsymbol{x}_1 \quad \cdots \quad \boldsymbol{x}_N]^\top \in \mathbb{R}^{N \times C}$ ($\boldsymbol{x}_i \in \mathbb{R}^C$) satisfies the conditions that $\boldsymbol{x}_i \neq 0$ for all $i \in [N]$ and $\boldsymbol{x}_i \nparallel \boldsymbol{x}_j$ for all $i \neq j \in [N]$. Let $\delta, \alpha, \beta > 0$. Then, there exists $R = O(N^6 \delta^{-3})$ such that for all $\nabla\widehat{\mathcal{L}}(\widehat{Y}^{(t-1)})$, with a probability of at least $1 - \delta$, the gradient descent algorithm finds $b^{(t)} \in \mathcal{B}^{(t)}$ such that the $t$-th weak learner $f^{(t)}$ satisfies the $(\alpha, \beta, -\nabla\widehat{\mathcal{L}}(\widehat{Y}^{(t-1)}))$-w.l.c.*

*Proof.* We define $H^\infty \in \mathbb{R}^{N \times N}$ by $H_{ij}^\infty := \mathbb{E}_{\boldsymbol{w} \sim \mathcal{N}(0,I)}[\boldsymbol{x}_i^\top \boldsymbol{x}_j \mathbf{1}\{\boldsymbol{w}^\top \boldsymbol{x}_i \geq 0\}\mathbf{1}\{\boldsymbol{w}^\top \boldsymbol{x}_j \geq 0\}]$. Let $\lambda_0$ be the lowest eigenvalue of $H^\infty$. Under the assumption, we know $\lambda_0 > 0$ by Theorem 3.1 of [6]. We train the parameter $\Theta_{\cdot c}$ using the dataset $((\boldsymbol{x}_1, -\alpha[\widehat{\mathcal{L}}(\widehat{Y}^{(t-1)})]_1), \ldots, (\boldsymbol{x}_N, -\alpha[\widehat{\mathcal{L}}(\widehat{Y}^{(t-1)})]_N))$. We denote the parameter of the MLP at the $k$-th iteration of the gradient descent by $\Theta^{(k)}$. We denote the output of the model $f_{\Theta^{(k)}}$ by $\boldsymbol{u}^{(k)} := (f_{\Theta^{(k)}}(\boldsymbol{x}_1), \ldots, f_{\Theta^{(k)}}(\boldsymbol{x}_N))^\top$. By [6, Theorem 4.1], with probability $1 - \delta$, we have

$$\|\boldsymbol{u}^{(k)} + \alpha\nabla\widehat{\mathcal{L}}(\widehat{Y}^{(t-1)})\|_2 \leq \left(1 - \frac{\eta\lambda_0}{2}\right)^k \|\boldsymbol{u}^{(k)} + \alpha\nabla\widehat{\mathcal{L}}(\widehat{Y}^{(t-1)})\|_2,$$

where $\eta = O\left(\frac{\lambda_0}{N^2}\right)$. Set

$$k := \log\left(\frac{\beta\|\nabla\widehat{\mathcal{L}}(\widehat{Y}^{(t-1)})\|_2}{\|\boldsymbol{u}^{(0)} + \alpha\nabla\widehat{\mathcal{L}}(\widehat{Y}^{(t-1)})\|_2 \vee 1}\right)\left(\log\left(1 - \frac{\eta\lambda_0}{2}\right)\right)^{-1}.$$

Table 2: Dataset specifications.

|          | #Node | #Edge | #Class ($K$) | Chance Rate |
|----------|-------|-------|--------------|-------------|
| Cora     | 2708  | 5429  | 6            | 30.2%       |
| CiteSeer | 3312  | 4732  | 7            | 21.1%       |
| PubMed   | 19717 | 44338 | 3            | 39.9%       |

Then, with probability $1 - \delta$, we have

$$\|\boldsymbol{u}^{(k)} + \alpha\nabla\widehat{\mathcal{L}}(\widehat{Y}^{(t-1)})\|_2 \leq \beta\|\nabla\widehat{\mathcal{L}}(\widehat{Y}^{(t-1)})\|_2,$$

which means $\boldsymbol{u}^{(k)}$ satisfies $(\alpha, \beta, -\nabla\widehat{\mathcal{L}}(\widehat{Y}^{(t-1)}))$-w.l.c. $\qquad\square$

**Remark 1.** *Reference [6] assumed that any feature vector $\boldsymbol{x}$ of the training data satisfies $\|\boldsymbol{x}\| = 1$. However, as commented in [6], we can loosen this condition as follows: there exists $c_{low}, c_{high} > 0$ such that any feature vector $\boldsymbol{x}$ satisfies $c_{low} \leq \|\boldsymbol{x}\| \leq c_{high}$.*

Although this instantiation provably satisfies the w.l.c. with high probability, its model complexity is extremely large because it has as many as $O(N^6)$ parameters. As we saw in Section 8, such a large model complexity is inevitable as long as the gradient can take arbitrary values.

## D   More Model Variants

**Input Injection**   A matrix $P \in \mathbb{R}^{N \times N}$ defines the aggregation model $\mathcal{G}_P := \{X \mapsto PX\}$ as we did in Section 7. Typical choices of $P$ are the (normalized) adjacency matrix, a GCN-like augmented normalized adjacency matrix, or the (normalized) graph Laplacian. If we choose $P' := \rho P + (1 - \rho)I_N$ for some $\rho \in [0, 1]$, it aggregates the representations in a lazy manner using $P$. In the similar spirit of [21, 32], there is another type of lazy aggregation that allows us to inject the information of unmixed features directly to the representations. Specifically, for $\rho \in [0, 1]$, we define the *input injection* model $\mathcal{G}_{\mathrm{II}}(\rho, P)$ by

$$\mathcal{G}_{\mathrm{II}}(\rho, P) := \{X \mapsto \rho P X + (1 - \rho)X^{(1)}\}.$$

On one hand, $\mathcal{G}_{\mathrm{II}}(\rho, P)$ equals $\mathcal{G}_P$ when $\rho = 1$. On the other hand, when $\rho = 0$, $\mathcal{G}_{\mathrm{II}}(\rho, P)$ ignores the effect of the representation mixing and employs the original features. We can identify $\mathcal{G}_{\mathrm{II}}(\rho, P)$ with $\{(X, X') \mapsto (\rho P X + (1 - \rho)X', X')\}$. Therefore, if we redefine a new input space as $\mathcal{X}' := \mathcal{X} \times \mathcal{X}$, which means that we double the input channel size, and preprocess features as $x_i \mapsto (x_i, x_i)$ for each $i \in V$, we can think the input injection model as an example of our model. For the augmented normalized adjacency matrix $\tilde{A}$, we refer to the model that uses $\mathcal{G}_{\mathrm{II}}(\rho, \tilde{A})$ as the set of aggregation functions $\mathcal{G}^{(t)}$ and the set of MLPs as $\mathcal{B}^{(t)}$ for all $t$ as GB-GNN-II. We conducted the same experiment as the one we did in Section 7 using GB-GNN-II. The result is reported in Section F.1.

## E   Details of Experiment Settings

### E.1   Dataset

We used the Cora [17, 24], CiteSeer [9, 24], and PubMed [24] datasets. Each dataset represents scientific papers as the nodes and citation relationships as the edges of a graph. For each paper, the genre of this paper is associated as a label. The task is to predict the genre of papers from word occurrences and the citation relationships. Table 2 shows the statistics of datasets. We obtained the preprocessed dataset from the code repository of [15] (`https://github.com/tkipf/gcn`) and split each dataset into train, validation, and test datasets in the same way as experiments in [15].

### E.2   Model

As shown in Table 4 in Section F.1, we have tested four base models: GB-GNN-Adj, GB-GNN-KTA, GB-GNN-II, and GB-GNN-SAMME.R. For each model, we consider three types of variants: (1) the

base model, (2) the model with fine tuning, and (3) models with different layer sizes ($L = 0, 2, 3, 4$), We have shown the result of variants (1) and (2) of GB-GNN-Adj and GB-GNN-KTA in the main paper. See Section F.1 for the results of other models.

### E.2.1 Node Aggregation Functions

For the aggregation functions $\mathcal{G}$, we used the matrix multiplication model with the augmented normalized adjacency matrix $\tilde{A}$ of the underlying graph $\mathcal{G}_{\tilde{A}}$ (for GB-GNN-Adj) and the KTA model $\mathcal{G}_{\text{KTA}}$ (for GB-GNN-KTA). We also employed the input injection model with the augmented normalized adjacency matrix $\mathcal{G}_{\text{II}}(\rho, \tilde{A})$ defined in Section D (for GB-GNN-II). For the KTA models, we used $\mathcal{G}_{\text{KTA}} := \{g : X \mapsto wX + \sum_{k=0}^{N_{\text{deg}}} w_k \tilde{A}^{2^k} X \mid w, w_k \in \mathbb{R}\}$. We treat weights $w$ and $w_k$'s in $\mathcal{G}_{\text{KTA}}$ as learnable parameters and the mixing parameter $\rho$ of $\mathcal{G}_{\text{II}}(\rho, \tilde{A})$ as a hyperparameter.

### E.2.2 Boosting Algorithms

We used two boosting algorithms SAMME and SAMME.R. SAMME is the default boosting algorithm and is applied to GB-GNN-Adj, GB-GNN-KTA, and GB-GNN-II. We used SAMME.R in combination with the matrix multiplication model $\mathcal{G}_{\tilde{A}}$ only (for GB-GNN-SAMME.R).

### E.2.3 Transformation Functions

For the transformation functions $\mathcal{B}$, we used MLPs with ReLU activation functions that have $L = 0, \ldots, 4$ hidden layers followed by the argmax operation. We showed results for the $L = 1$ model in the main paper. See Section F.1 for other models.

The SAMME algorithm assumes that each weak learner outputs one of label categories, while SAMME.R assumes that the probability distribution over the set of categorical labels. Therefore, we added the argmax operation to the MLPs when we used SAMME and the softmax operation to the MLPs when SAMME.R,

We only imposed soft restrictions on MLPs in the models using regularization methods such as Dropout and weight decay. This is different from the MLP model defined in Section 5 in the main paper, which hard-thresholded the norms of weights and bias.

We treat weights in the MLP as trainable parameters and treat architectural parameters (e.g., unit size) other than the layer size $L$ as hyperparameters (see Table 3 for the complete hyperparameters).

### E.3 Training

We used the SAMME or SAMME.R algorithm to train the model. At the $t$-th iteration, we give $X^{(t-1)}$ and $Y$ as a set of feature vectors and labels, respectively to the model. We picked $B$ training sample points randomly and trained the transformation functions $\mathcal{B}^{(t)}$ using them. We used a gradient-based optimization algorithm to minimize the cross entropy between the prediction of the weak learner and the ground truth labels. We initialized the model (i.e., MLP) using the default initialization method implemented in PyTorch.

For the aggregation model $\mathcal{G}^{(t)}$, if it does not have a learnable parameter, that is, if $\mathcal{G}^{(t)}$ consists of a single function, we just applied the function to convert $X^{(t-1)}$ into $X^{(t)}$. For the KTA model, which has learnable parameters $w$ and $w_k$'s, we trained the model $g$ using a gradient-based optimization to maximize the correlation between gram matrices created from transformed features $g(X^{(t-1)})$ and labels $Y$. The correlation is defined as follows:

$$\frac{\langle \mathcal{K}[g(X^{(t-1)})], \mathcal{K}[Y] \rangle}{\|\mathcal{K}[g(X^{(t-1)})]\|_{\text{F}} \|\mathcal{K}[Y]\|_{\text{F}}},$$

where $\mathcal{K}$ is the operator that takes the outer product of training sample points defined in Section 6. We initialized weights $w$ and $w_k$'s with 1.

After the training using boosting algorithm, we optionally trained the whole model as fine-tuning. When we used SAMME, we replaced the argmax operation in the transformation functions $\mathcal{B}$ with the softmax function along class labels to make the model differentiable. When we used SAMME.R, we did not change the same architecture in the training and fine tuning phases. We trained the whole

Table 3: Hyperparameters of experiments in Section 7. $X \sim \text{LogUnif}[a, b]$ means the random variable $\log_{10} X$ obeys the uniform distribution over $[a, b]$. $(*)$ For KTA + Fine Tuning setting, we reduce the number of weak learners to 40 due to GPU memory constraints. $(**)$ Learning rate corresponds to $\alpha$ when Optimization algorithm is Adam [14].

| Category | Name | Value |
|---|---|---|
| Boosting | #Weak learners | $\{1, 2, \ldots, 100 \ (40^{(*)})\}$ |
| | Minibatch size $B$ | $\{1, 2, \ldots, |V_{\text{train}}|\}$ |
| | Clipping value | $\text{LogUnif}[-10, -5]$ |
| Model | Epoch | $\{10, 20, \ldots, 100\}$ |
| | Optimization algorithm | $\{\text{SGD}, \text{Adam}, \text{RMSProp}\}$ |
| | Learning rate$^{(**)}$ | $\text{LogUnif}[-5, -1]$ |
| | Momentum | $\text{LogUnif}[-10, -1]$ |
| | Weight decay | $\text{LogUnif}[-10, -1]$ |
| | Unit size | $\{10, 11, \ldots, 200\}$ |
| | Dropout | $\{\text{ON(ratio=0.5)}, \text{OFF}\}$ |
| Input Injection | Mixing ratio $\rho$ | $\text{Unif}[0, 1]$ |
| Kernel Target Alignment | Epoch | $\{5, 6, \ldots, 30\}$ |
| | Optimization algorithm | $\{\text{SGD}, \text{Adam}, \text{RMSProp}\}$ |
| | Learning rate$^{(**)}$ | $\text{LogUnif}[-5, -1]$ |
| | Degree $N_{\text{deg}}$ | 3 |
| Fine Tuning | Epoch | $\{1, 2, \ldots, 100\}$ |
| | Optimization algorithm | $\{\text{SGD}, \text{Adam}, \text{RMSProp}\}$ |
| | Learning rate$^{(**)}$ | $\text{LogUnif}[-5, -1]$ |
| | Momentum | $\text{LogUnif}[-10, -1]$ |
| | Weight decay | $\text{LogUnif}[-10, -1]$ |

model in an end-to-end manner using a gradient-based optimization algorithm to minimize the cross entropy between the prediction of the model and the ground truth label.

### E.4 Evaluation

We split the dataset into training, validation, and test datasets. For each hyperparameter, we trained a model using the training dataset and evaluated it using the validation dataset. We defined the performance of a set of hyperparameters as the accuracy on the validation dataset at the iteration that maximizes the validation accuracy. If a model has a fine-tuning phase, we used the accuracy after the fine-tuning as the performance. We chose the set of hyperparameters that maximizes the performance using a hyperparameter optimization algorithm. We employed Tree-structured Parzen Estimator [3] and for hyperparameter optimization and the median stopping rule implemented in Optuna for pruning unpromising sets of hyperparameters. Table 3 shows the set of hyperparameters. We define the final performance of the model as the accuracy on the test dataset attained by the optimized set of hyperparameters.

For each pair of the dataset and the model, we ran the above evaluation ten times and computed the mean and standard deviation of the performance.

### E.5 Implementation and Computational Resources

Experimental code is written in Python3. We used PyTorch [22] and Ignite for the implementation and training of models, Optuna [1] for the hyperparameter optimization, NetworkX [11] for preprocessing graph objects, and SciPy [28] for miscellaneous machine learning operations. We ran each experiment on a docker image (OS: Ubuntu18.04) built on a cluster. The image has two CPUs and single GPGPUs (NVIDIA Tesla V100).

Table 4: Accuracy of node classification tasks on citation networks. $L$ denotes the number of hidden layers. Numbers are (mean) $\pm$ (standard deviation) of ten runs. ($*$) All runs failed due to GPU memory errors.

| | $L$ | Cora | CiteSeer | PubMed |
|---|---|---|---|---|
| GB-GNN-Adj | 0 | $79.4 \pm 1.9$ | $70.3 \pm 0.5$ | $78.8 \pm 0.7$ |
| | 1 | $79.9 \pm 0.8$ | $70.5 \pm 0.8$ | $79.4 \pm 0.2$ |
| | 2 | $79.9 \pm 1.3$ | $68.5 \pm 1.3$ | $78.9 \pm 0.6$ |
| | 3 | $77.4 \pm 0.8$ | $64.4 \pm 1.5$ | $78.0 \pm 0.5$ |
| | 4 | $75.6 \pm 2.6$ | $60.7 \pm 1.7$ | $77.9 \pm 0.6$ |
| GB-GNN-Adj. + Fine Tuning | 1 | $80.4 \pm 0.8$ | $70.8 \pm 0.8$ | $79.0 \pm 0.5$ |
| GB-GNN-KTA | 0 | $80.0 \pm 0.8$ | $70.0 \pm 1.8$ | $79.4 \pm 0.1$ |
| | 1 | $80.9 \pm 0.9$ | $73.1 \pm 1.1$ | $79.1 \pm 0.4$ |
| | 2 | $79.8 \pm 1.3$ | $68.8 \pm 1.1$ | $79.1 \pm 0.4$ |
| | 3 | $78.5 \pm 0.9$ | $65.2 \pm 1.5$ | $78.4 \pm 0.8$ |
| | 4 | $76.0 \pm 2.5$ | $65.6 \pm 1.7$ | $78.0 \pm 0.7$ |
| GB-GNN-KTA + Fine Tuning | 1 | $82.3 \pm 1.1$ | $70.8 \pm 1.0$ | N.A.$^{(*)}$ |
| GB-GNN-II | 0 | $79.2 \pm 1.3$ | $71.4 \pm 0.3$ | $79.3 \pm 0.5$ |
| | 1 | $79.8 \pm 1.3$ | $71.3 \pm 0.5$ | $79.4 \pm 0.3$ |
| | 2 | $79.9 \pm 0.8$ | $69.8 \pm 1.1$ | $79.3 \pm 0.3$ |
| | 3 | $78.7 \pm 1.7$ | $66.7 \pm 1.9$ | $79.2 \pm 0.6$ |
| | 4 | $75.4 \pm 2.0$ | $65.1 \pm 2.5$ | $78.6 \pm 0.7$ |
| GB-GNN-II + Fine Tuning | 1 | $80.8 \pm 1.3$ | $70.8 \pm 0.9$ | $79.2 \pm 0.8$ |
| GB-GNN-SAMME.R | 0 | $81.0 \pm 0.8$ | $70.4 \pm 0.7$ | $78.9 \pm 0.3$ |
| | 1 | $82.2 \pm 1.2$ | $71.6 \pm 0.5$ | $78.8 \pm 0.3$ |
| | 2 | $80.5 \pm 0.8$ | $67.4 \pm 1.1$ | $78.9 \pm 0.3$ |
| | 3 | $79.6 \pm 1.1$ | $64.4 \pm 1.4$ | $78.8 \pm 0.5$ |
| | 4 | $78.9 \pm 1.8$ | $64.6 \pm 1.3$ | $78.1 \pm 0.6$ |
| GB-GNN-SAMME.R + Fine Tuning | 1 | $82.1 \pm 1.0$ | $71.3 \pm 0.8$ | $79.4 \pm 0.4$ |

## F  Additional Experiment Results

### F.1  More Results for Model Variants

Table 4 shows the result of the prediction accuracies of models that use MLPs with various layer size $L$ as transformation functions $\mathcal{B}^{(t)}$. It also shows the results for the input injection model (GB-GNN-II) we have introduced in Section D and the SAMME.R model (GB-GNN-SAMME.R) in Section E.2. Figures 3–5 show the transition of the training loss for $L = 0, \ldots, 4$ (Figure 3: Cora, Figure 4: CiteSeer, Figure 5: PubMed). Figures 6–8 show the transition of the training loss for $L = 0, \ldots, 4$ (Figure 6: Cora, Figure 7: CiteSeer, Figure 8: PubMed). Figures 9–11 show the transition of the cosine values between the negative gradient $-\nabla \widehat{\mathcal{L}}(\widehat{Y}^{(t-1)})$ and the weak learner $f^{(t)}$ at the $t$-th iteration for models that has $L = 0$ to $4$ layers (Figure 9: Cora, Figure 10: CiteSeer, Figure 11: PubMed).

### F.2  Performance Comparison with Existing GNN Models

Table 5 shows the accuracies of node prediction tasks on citation networks for various GNN models. We borrowed the results of the official repository of Deep Graph Library (GDL) [29] (`https://github.com/dmlc/dgl`), a package for deep learning on graphs.

Figure 3: Train loss transition for the Cora dataset.

Figure 4: Train loss transition for the CiteSeer dataset.

Figure 5: Train loss transition for the PubMed dataset.

Figure 6: Test loss transition for the Cora dataset.

Figure 7: Test loss transition for the CiteSeer dataset.

Figure 8: Test loss transition for the PubMed dataset.

Figure 9: The similarity $\cos\theta^{(t)}$ between weak learners $f^{(t)}$ and the gradient $\nabla\widehat{\mathcal{L}}$ of the training loss for the Cora dataset.

Figure 10: The similarity $\cos\theta^{(t)}$ between weak learners $f^{(t)}$ and the gradient $\nabla\widehat{\mathcal{L}}$ of the training loss for the CiteSeer dataset.

Figure 11: The similarity $\cos\theta^{(t)}$ between weak learners $f^{(t)}$ and the gradient $\nabla\widehat{\mathcal{L}}$ of the training loss for the PubMed dataset.

Table 5: Comparison of accuracy of GNN models. Created from the official repository of DGL as of May 23rd, 2020. Adj.: Matrix multiplication model $\mathcal{G}_{\tilde{A}}$ by the normalized adjacency matrix $\tilde{A}$. KTA: Kernel target alignment model $\mathcal{G}_{\mathrm{KTA}}$. II: Input injection model $\mathcal{G}_{\mathrm{II}}$. FT: Fine Tuning. Paper: accuracies are cited from the paper in the Ref. column. DGL: accuracies are cited from the official implementation of DGL. ($*$) Not available due to GPU memory errors. ($**$) Not available from the DGL repository.

| Model | | Ref. | Source | Framework | Cora | Citeseer | Pubmed |
|---|---|---|---|---|---|---|---|
| GB-GNN | Adj | – | – | PyTorch | 79.9 | 70.5 | 79.4 |
| | Adj + FT | | | | 80.4 | 70.7 | 79.0 |
| | KTA | | | | 80.9 | 73.1 | 79.4 |
| | KTA + FT | | | | 82.3 | 70.8 | N.A.$^{(*)}$ |
| | II | | | | 79.8 | 71.4 | 79.4 |
| | II + FT | | | | 80.8 | 70.8 | 79.2 |
| | SAMME.R | | | | 82.2 | 71.6 | 78.8 |
| | SAMME.R + FT | | | | 82.1 | 71.3 | 79.4 |
| SGC | | [30] | Paper | – | 83.0 | 72.5 | 79.0 |
| | | | DGL | PyTorch | 84.2 | 70.9 | 78.5 |
| GCN | | [15] | Paper | – | 81.5 | 70.3 | 79.0 |
| | | | DGL | PyTorch | 81.0 | 70.2 | 78.0 |
| | | | | TensorFlow | 81.0 | 70.7 | 79.2 |
| TAGCN | | [4] | Paper | – | 83.3 | 71.4 | 79.4 |
| | | | DGL | PyTorch | 81.2 | 71.5 | 79.4 |
| | | | | MXNet | 82.0 | 70.2 | 79.8 |
| DGI | | [26] | Paper | – | 82.3 | 71.8 | 76.8 |
| | | | DGL | PyTorch | 81.6 | 69.4 | 76.1 |
| | | | | TensorFlow | 81.6 | 70.2 | 77.2 |
| GraphSAGE | | [12] | DGL | PyTorch | 83.3 | 71.1 | 78.3 |
| | | | | MXNet | 81.7 | 69.9 | 79.0 |
| APPNP | | [16] | Paper | – | 85.0 | 75.7 | 79.7 |
| | | | DGL | PyTorch | 83.7 | 71.5 | 79.3 |
| | | | | MXNet | 83.7 | 71.3 | 79.8 |
| GAT | | [25] | Paper | – | 83.0 | 72.5 | 79.0 |
| | | | DGL | PyTorch | 84.0 | 70.9 | 78.6 |
| | | | | TensorFlow | 84.2 | 70.9 | 78.5 |
| MoNet | | [19] | DGL | PyTorch | 81.6 | N.A.$^{(**)}$ | 76.3 |
| | | | | MXNet | 81.4 | N.A.$^{(**)}$ | 74.8 |

## Footnotes

[1] We are not aware the standard notion used to tell apart the complexities defined in Definitions 2 and 4. The notion of *(un)symmetrized* is specific to this paper.