[Reviews · NeurIPS 2020]

Review 1

Summary and Contributions: This paper analyzes a boosting algorithm that is related to multi-scale GNN, for optimization and generalization bounds. It also provides empirical observations for the boosting algorithm.

Strengths: The claims are sound, the paper is not too incremental, and relevant to the NeurIPS community.

Weaknesses: The claims are sound, but somewhat trivial given previous results. The proofs are simply the combinations of standard known results. So, this paper does not provide a new proof technique or insights in terms of theory. But, it applies that to an interesting question about multi-scale GNN, which is good.

Correctness: The claims seem to be correct overall. The claim on "test error bound monotonically decreasing with depth" should be deleted for sure. This is not supported by the results here. This claim requires the several assumptions implicitly made in this paper that are not validated and would be invalidated in the future. First, the test error bound in theorem 2 contains D^(t), which contains 2^L factor, which increases exponentially as the depth increases. There are also other terms that can increase as depth increase. Therefore, "the test error bound monotonically decreasing with depth" is not valid, and I suspect the opposite: this test error bound indeed increases with depth for practical cases. But, I understand that this seems to stem from discrepancy in terminology at first glance. Depth in this paper means depth in aggregation functions over t, instead of depth for classification networks L. But, depth in aggregation functions do not contain any nonlinear activation, which is the reason why this bound does not contain 2^T factor or other factor that grows exponentially in t in the test error bound. If we use non-linearity in-between aggregation functions G, we have the same issue for depth in aggregation functions over t too. Therefore, this is not just due to the discrepancy in terminology, but fundamental in proof.

Clarity: The paper is well written. It also provides previous related papers well.

Relation to Prior Work: This paper clearly discusses how this work differs from previous contributions.

Reproducibility: Yes

Additional Feedback: I read the author feedback. It answers my question well and was consistent with what I assumed in the original review. Therefore, I remain my positive evaluation. In my understanding, in standard multi-scale GNN, there are nonlinear activation in-between aggregation functions G. In this paper, there is no nonlinear activation in-between aggregation functions G. Nonlinear activation is only in B. Therefore, "graph" part G is always linear. Is there such multi-scale GNN in the literature? Why do you think this "linear version" works better than standard multi-scale GNN with non-linearity in-between aggregations? Is there any reference or additional empirical study to demonstrate the effect of "linear version vs nonlinear version"? The answer to these questions affect my final evaluation a lot, since this determines how relevant this paper is for GNN. After author response on this, I may decrease or increase my evaluation a lot (my current evaluation is very tentative). Theorem 1 is trivial to me given Assumption 1, even without any previous papers. But, I have worked on this field, and what I thought trivial turned out to be not trivial to others in many times previously. Theorem 2 (with the correction in appendix) is more interesting to me, but this also follows standard proof of Rademacher complexity bounds. This bound can increase exponentially as depth increase, as I mentioned above. The trade-off issue mentioned after Theorem 2 is not resolved in this paper. But, it is good to discuss the limitation explicitly, unlike many other papers.


Review 2

Summary and Contributions: The paper presents a novel boosting style method to optimize graph NNs and tries to pull together various pieces from learning theory to shed light on the optimization and generalization accuracy in GNNs, in particular as it relates to a phenomenon called "over-smoothing" in the DNN community.

Strengths: The paper digests quite some literature and results, which it tries to adapt to the chosen setting.

Weaknesses: I find the paper extremely hard to read. I can't state that I feel comfortable, I have understood the main claims and results. There is very little empirical evidence or clean argumentation that would motivate the reader to follows the suggested path that combines "muddy" fundings from GNNs with established learning theory. As it stands, I find the paper needs a major revision to work out the actual claims and results.

Correctness: (Unable to check due to complexity of write-up.)

Clarity: No, surely not. See above.

Relation to Prior Work: A lot of work is cited, but I still feel there is a lack of clarity in putting this work into the context of other work in GNNs.

Reproducibility: Yes

Additional Feedback:


Review 3

Summary and Contributions: This paper proposed GB-GNN, a multi-scale graph neural network with the boosting theory to deal with the over-smoothing problem. On top of the generalization gap, the authors also studied the optimization guarantee under the weak learning condition. The upper bound of test error is derived under the special case of MLPs as the transformation function and GCNs as the aggregation function. Both theoretical analysis and experimental results indicate an improved performance on test error when deepening the model. In terms of the prediction accuracy, the framework (with several variants) has been implemented on the standard node classification datasets and performs comparable results to the conventional GNNs.

Strengths: The proposed idea combines the idea of multi-scale GNNs and boosting theory in ensemble methods. Under the condition of weak learnability, the proposed model can derive the optimization and generalization guarantees. With an increased depth (the training iteration), the model's generalization bound is monotonically decreasing and thus avoid the over-smoothing problem in general GNN frameworks. Detailed proofs are formulated to fill the insufficient discussion in literature, and empirical evidence is provided with three node classification benchmark datasets, which suggests the comparable performance of the underlying GB-GNN models.

Weaknesses: 1. According to the notations in Equation (3), I believe that the model fits a separate transformation function $b^{(t)}$ by an L-layer MLP at each iteration~$t$. If that is the case, the weight to be estimated and the number of hyper-parameters to be tuned could be large, especially when increasing the iteration number and layers. I would expect a small section concerning the efficiency of the algorithm and the performance sensitivity to the parameter settings. 2. According to Section 5, the performance of this multi-scale ensemble model relies heavily on the depth of the model. However, the wlc condition could easily break when it goes too deep. That is, for a larger dataset with a more complex structure, the model could fail before learning sufficient information. 3. Although theoretically appealing, the GB-GNN model does not outperform many competitors in experiments, according to Table 5 of Section G. Even for the three standard datasets that are baselines for node classification, yet still one of the models cannot run properly. Given this, the model's capability on large datasets is questionable.

Correctness: The proof seems correct.

Clarity: Generally, the paper is pretty readable; however, the notation could be improved in a couple of places. 1. It would be easier to follow if all the notations are properly defined right beside the equations. For example, the definitions of $\tilde{B}^{(t)}$ at line 198? The constant $\tilde{C}^{(t)}$ in Equation 4 are not clear. I can see it is a positive number and an upper bound constraint for $||W_C||_1$, but what's the meaning of it? What's the difference between $C^{(t)}$ (for example, in line 668 - 669) and $\tilde{C}^{(t)}$? 2. It is hard to understand Equation 4 fully. I assume the model uses one layer of inactivated GCN in each iteration with the same weight set $W$ and adjacency matrix $\tilde{P}$ to transform $X^{(t-1)}$ to $X^{(t)}$. Please correct me if I am wrong; otherwise, it would be clearer to add superscript $(t-1)$ and $(t)$ to X.

Relation to Prior Work: The authors stated in Supplementary that several existing works in a similar field. However, their analysis either restricted to a specific model, or they only considered the generalization gap but left the optimization guarantee aside. This work extends the idea in [67] to multi-layered and multi-scale GNNs, and include optimization analysis on top of [25].

Reproducibility: Yes

Additional Feedback: The w.l.c params ($\alpha_t, \beta_t$) are quite confusion to me. According to Algorithm 1, they are inputs in each iteration. However, they are not hyper-parameters to tune with empirical loss (at least not shown in Table 3 of Section E4), and they are not part of the loss functions. I wonder how these parameters influence the mode performance, and how are they valued? 2. In Equation 4, I wonder why applying $\tilde{P}$ could exclude neighbouring information (line 42)? 3. According to Section G1 and Section G2, especially for the Cora dataset, one hidden layer MLPs significantly outperforms other 0-4 hidden layer choices, for both training and test losses. The models also perform more stable in terms of the accuracy score. Does it indicate a `weaker' learner is favourable than deep `stronger' learners? What are the possible explanations of that?

[Author Response · NeurIPS 2020]

**[ID 3407]** We thank reviewers for their efforts and giving us valuable comments. *Q.N[-M]* is a response to Reviewer N.

*Q.1-1 It is unclear whether "depth" refers to $L$ or $T$. In either case, the claim "test error bound monotonically decreasing with depth" is inappropriate.* A. The depth (e.g., in L.42) meant $T$. We considered the dependence on $T$ rather than $L$ because we are interested in the over-smoothing caused by node aggregations. [Depth=$L$ case] It is true that the bound can exponentially depend on $L$. Since this problem occurs in inductive MLPs, too, many studies derived generalization bounds that avoid this exponential dependency [Arora+ICML18; Nagarajan+ICLR19; Wei+NeuIPS19]. We can incorporate them to obtain tighter bounds (c.f., Remark 2). [Depth=$T$ case] See Q.1-2. We admit that the expression L.46–47 was confusing because it is not clear that depth means $T$, and this bound needs not only w.l.c. but also the condition in L.213. We will address this problem in the updated version.

*Q.1-2 Why did you think of a liner model as $\mathcal{G}^{(t)}$? What is the superiority of linear $\mathcal{G}^{(t)}$ over non-linear one?* A. GNNs that consist of linear node aggregations and non-linear MLPs is one trend in the GNN research. For example, SGC [69], gfNN [NT+19(arXiv:1905.09550)], and APPNP [41] are such examples. Theoretically, [NT+19] and [51] claimed that non-linearity between aggregations is not essential for predictive performance. So, we believe that such models are worth investigating. [Superiority] Adding non-linearity changes two things. First, $\|P^{(t)}X\|_F$ in (5) is replaced with $\prod_{s=2}^{t}\|\tilde{P}^{(s)}\|_{op}\|X\|_F$. It makes the interpretation of L.222–234 impossible, and the bound looser, essentially because the bound loses the information of eigenvectors. Second, the bound of Rademacher complexity of $\mathcal{F}^{(t)}$ is multiplied by $2^t$ (not $2^T$). It changes the condition in L.213 to a stricter one $\alpha_t^{-1}2^t D^{(t)}\prod_{s=2}^{t}\|\tilde{P}^{(s)}\|_{op} = O(\tilde{\varepsilon}^t)$. Nevertheless, we do not have a definitive answer whether "linear" GNNs are truly superior to "non-linear" ones. We may be able to use techniques similar to [51] for the first problem. Refined analyses could eliminate the $2^t$ term for the second problem.

*Q.2 The paper is extremely hard to read because there is very little empirical evidence or clean argumentation that would motivate the reader to follows the suggested path.* A. Sec.1 Par.2 and Sec.2 Par.2 correspond to the empirical superiority of multi-scale GNNs, and Sec.1 Par.3 and Sec.2 Par.3 to the motivation for using the boosting interpretation and learning theory. We are sorry that the paper has unclear points. However, we believe our paper is well-written, as other reviewers evaluated the clarity of our paper. We want the reviewer to reread it and reconsider the evaluation.

*Q.3-1 The number of weights and hyperparameters could be large for large $T$ and $L$ [3-1].* A. For hyperparameters, we used the same hyperparameter set for every weak learner in experiments. The number of hyperparameters is indep. of $T$ for this setting. The accuracy is as high as existing methods, even though such a simplification. For weights, $L = 1$ was enough for empirical performance. The number of weights in that model is comparable to a standard $3T$-layered MLP.

*Q.3-2 The GB-GNN model does not outperform many competitors in experiments [3-2]. Besides, one of the models fails to run properly, even for the standard datasets [3-3].* A. We put importance on the consistency of theoretical and empirical behaviors, rather than achieving SOTA performances. Observing that many of the SOTA methods have little performance difference in benchmark datasets, we might almost reach the performance limit. Such quality issues of standard benchmarks are a major challenge in the GNN community and motivate recent benchmarking researches (e.g., Open Graph Benchmark, [Errica+ICLR20], and [Dwivedi+20(arXiv2003.00982)]). Considering the current situation and the theoretical nature of this paper, we think that accuracy comparable to existing methods is sufficient to guarantee our method's correctness. Regarding the failure, there are two reasons. First, our implementation naively processes all nodes at once. Second, since we train the fine-tuning model in an end-to-end manner, it uses memory proportional to $T$. These problems are not specific to our model but common to end-to-end deep GNN models. We can solve them by mini-batching. Also, we note that when training without fine-tuning, memory usage is constant w.r.t. $T$, since we do not have to retain intermediate weights and outputs. This memory-efficiency is an advantage of the boosting algorithm.

*Q.3-3 The w.l.c. params are confusing. How do they influence the model performance, and how are they valued [8-1]?* A. Algorithm 1 pre-determines w.l.c. params and train weak learners until the w.l.c. condition is satisfied. However, it is practically more convenient to train weak learners and determine w.l.c. params that they satisfy *a posteriori*. So, we did not evaluate how the change in w.l.c. params affect the performance empirically. Still, we can control w.l.c. params indirectly by the complexity of weak learners (e.g., width, layer size) and check the satisfiability of w.l.c. (c.f., Fig.2).

*Q.3-4 In Equation 4, why could applying $\tilde{P}$ exclude neighbouring information (L.42) [8-2]?* A. The *representation* in L.42 is the input for $b^{(t)}$ (i.e., not $X$ but $P^{(t)}X$). What we intended was that by assuming the model in Sec.5, we could evaluate the model complexity using inductive models (c.f., Remark 2). We are sorry for the confusion.

*Q.3-5 Do the experiment results indicate that a "weaker" learner is favourable than deep "stronger" ones [8-3]?* A. We think both "weak" and "strong" learners can be problematic. If we use "strong" learners, we have a risk of over-fitting, especially when $T$ is small. If we use "weak" learners, we can not reduce training loss and cannot make up for it in later iterations using over-smoothed representations. The present results suggest to balance between the two situations.

*Q.3-6 Clarification of notations [5-1, 5-2].* A. $\tilde{B}^{(t)}$ is a user-defined constant in parallel to $C_l$'s and $L$. $\tilde{C}^{(t)}$ is defined in L.201. In equation 4, different from the reviewer's comment, we use different weight matrices for each iteration.

[Meta-Review · NeurIPS 2020]

The paper considers multi-scale GNNs which have been shown to address over-smoothing issues with standard GNNs, and establishes optimization and generalization guarantees from the perspective of gradient boosting. The paper also suggests GB-GNN with linear transformations, and illustrates that the model can be competitive with the state-of-the-art. Most reviewers felt that the work presents a unique perspective to the performance of multi-scale GNNs. There are some concerns regarding the work - the technical results follow from assumptions and existing results on transductive learning, so there is limited core technical novelty. The work analyzes linear transformations which is different from nonlinear transformations often used in practice. It is unclear if the assumptions needed for the analysis are valid, or how to verify them.